# Unifying Lower Bounds on Prediction Dimension of Consistent Convex Surrogates

**Jessie Finocchiaro**
University of Colorado Boulder
jessica.finocchiaro@colorado.edu

**Rafael Frongillo**
University of Colorado Boulder
raf@colorado.edu

**Bo Waggoner**
University of Colorado Boulder
bwag@colorado.edu

## Abstract

The convex consistency dimension of a supervised learning task is the lowest prediction dimension $d$ such that there exists a convex surrogate $L : \mathbb{R}^d \times \mathcal{Y} \to \mathbb{R}$ that is consistent for the given task. We present a new tool based on property elicitation, $d$-flats, for lower-bounding convex consistency dimension. This tool unifies approaches from a variety of domains, including continuous and discrete prediction problems. We use $d$-flats to obtain a new lower bound on the convex consistency dimension of risk measures, resolving an open question due to Frongillo and Kash (NeurIPS 2015). In discrete prediction settings, we show that the $d$-flats approach recovers and even tightens previous lower bounds using feasible subspace dimension.

## 1 Introduction

A loss function is called a *surrogate* when it is used to solve a related, but not identical, "target" problem of interest. Selecting a hypothesis by minimizing surrogate risk is one of the most widespread techniques in supervised machine learning. There are two main reasons why a surrogate loss is necessary: (I) the target problem is to minimize a loss, the *target loss*, that does not satisfy some desiderata such as continuity or convexity; or (II) the target problem is to estimate some *target statistic* and some associated surrogate loss is required to do so, as in many continuous estimation problems. In both settings, a key criteria for choosing a surrogate loss is *consistency*, a precursor to excess risk bounds and convergence rates. Roughly speaking, consistency means that minimizing surrogate risk corresponds to solving the target problem of interest, i.e. in (I) the target risk is also minimized, or in (II) the continuous prediction approaches the true conditional statistic.

Despite the ubiquity of surrogate losses, we lack general frameworks to design and analyze consistent surrogates. This state of affairs is especially dire when one seeks low *prediction dimension*, the dimension of the surrogate prediction domain. For example, in multiclass classification with $n$ labels, the prediction domain might be $\mathbb{R}^n$. In many type (I) settings, such as structured prediction and extreme classification, the prediction dimension of any convex and consistent surrogate often becomes intractably large, forcing one to sacrifice consistency for computational efficiency. To understand whether this sacrifice is necessary, recent work developed tools like the feasible subspace dimension to lower bound the prediction dimension of any consistent convex surrogate [33]. Challenges of type (II) include estimating risk measures such as conditional value at risk (CVaR), with applications in financial regulation, robust engineering design, and algorithmic fairness [1, 14, 35, 42]. Risk measures

35th Conference on Neural Information Processing Systems (NeurIPS 2021).

are not elicitable, meaning they cannot be specified via a target loss, and thus we seek a surrogate loss of low (or at least finite) prediction dimension. Recent work [15, 19, 20] gives prediction dimension bounds for some of these risk measures, but without the requirement that the surrogate be convex; bounds for convex surrogates are left as a major open question.

We present a new tool, *d-flats*, which unifies existing techniques to bound the convex consistency dimension in both settings above. Using this tool, we resolve the above open question for type (II), giving the first prediction dimension bounds for risk measures with respect to convex surrogates. We also resolve a similar open question for the mode and modal interval, posed by Dearborn and Frongillo [10]. In settings of type (I), *d*-flats recover and tighten the feasible subspace dimension result of Ramaswamy and Agarwal [33]. Our framework rests on *property elicitation*, a weaker and simpler condition than calibration, as a way to understand consistency across a wide variety of domains.

**The "four quadrants" of problem types.**   Above, we discuss a significant divergence in previous frameworks: constructing a surrogate given a *target loss* versus a *target statistic*. In addition to the two possible targets, we may have one of two domains: a *discrete* (i.e. finite) target prediction space, like a classification problem, or a *continuous* one, like a regression or point estimation problem. We informally refer to the four resulting cases—target loss vs. target statistic, and discrete vs. continuous predictions—as the "four quadrants" of supervised learning problems, shown in Table 1. In the context of these quadrants, Figure 1 gives a roadmap of our main results.

**Literature on consistency and calibration.**   We focus on surrogate losses $L : \mathbb{R}^d \times \mathcal{Y} \to \mathbb{R}$ that are consistent, roughly meaning that minimizing $L$-loss corresponds to solving the target problem of interest.

We give informal definitions of consistency in § 2.2, with formal definitions in § A.

When given a *target loss* $\ell$, we roughly define $L$ to be consistent if minimizing $L$, and applying a link function, minimizes $\ell$ [33, 39, 41, 44]. When given instead a target statistic such as the conditional quantile or variance, we introduce a notion of consistency in line

|  | *Target loss* | *Target statistic* |
|---|---|---|
| *Discrete prediction* | **Q1**, e.g. classification | **Q2**, e.g. hierarchical classification |
| *Continuous estimation* | **Q3**, e.g. least-squares regression | **Q4**, e.g. variance estimation |

Table 1: The four quadrants of problem types, with an example for each as discussed in § 3.1.

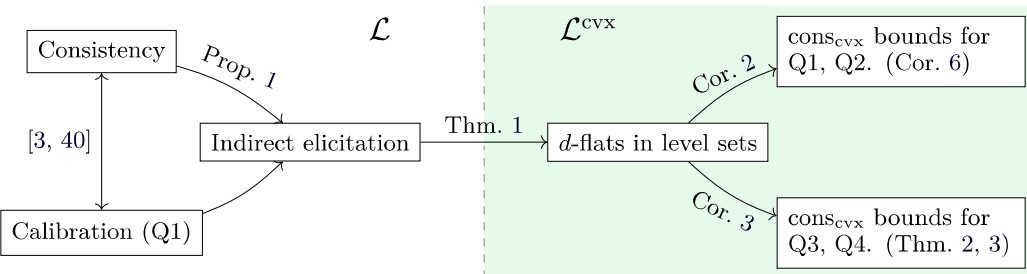

Figure 1: Flow and implications of our results. Compared to calibration, we suggest indirect elicitation as a simpler but almost-as-powerful necessary condition for consistency. In particular, we obtain a testable condition (Theorem 1), based on *d*-flats, for the existence of a *d*-dimensional consistent convex surrogate. This condition recovers and strengthens existing calibration-based results for Q1, while simultaneously applying to other quadrants. We illustrate the breadth and power of *d*-flats by resolving two open questions for Q3 and Q4 in § 4.

with classical statistics [11, 21, 36]. Here we define $L$ to be consistent if minimizing $L$ and applying a link function yields predictions converging to the correct statistic value. The key observation which underpins our approach is that consistency for target losses is a special case of consistency for target statistics (Lemma 1). Therefore, property elicitation—which studies the exact minimizers of loss functions—allows us to give general lower bounds on prediction dimension of any convex surrogates corresponding to a target task; these bounds apply across all four quadrants. See § 2.3 for other prior work on notions of prediction dimension.

As definitions of consistency are difficult to apply directly, the literature often focuses on a weaker condition called *calibration*, which only applies when given a target loss, e.g. Quadrants 1 and 3. Particularly, several authors [3, 28, 33, 41, 44] show the equivalence of consistency and calibration in Quadrant 1. We discuss the additional relationship of elicitation and calibration in § C, and re-derive Proposition 1 via calibration.

## 2   Setting

In supervised learning, data is drawn from a distribution $D$ over the space $\mathcal{X} \times \mathcal{Y}$ and the goal is to produce a hypothesis $f : \mathcal{X} \to \mathcal{R}$. Here $\mathcal{X}$ is the *feature space*, $\mathcal{Y}$ the *label space*, and $\mathcal{R}$ the *report* or *prediction space*, possibly different from $\mathcal{Y}$. For example, in ranking problems, $\mathcal{R}$ may be all $|\mathcal{Y}|!$ permutations over the $|\mathcal{Y}|$ labels forming $\mathcal{Y}$. We focus on surrogate losses, target problems, and their relationships to conditional distributions $p := D_x = \Pr[Y|X = x]$ over $\mathcal{Y}$ given some $x \in X$. We can often abstract away $x$, working directly with a set of (conditional) distributions over outcomes $\mathcal{P} \subseteq \Delta_\mathcal{Y}$, where $\Delta_\mathcal{Y} := \{p \in \mathbb{R}_+^{|\mathcal{Y}|} \mid \|p\| = 1\}$ is the probability simplex over labels. We then write e.g. $\mathbb{E}_p \ell(r, Y)$ to mean the expected loss of prediction $r \in \mathcal{R}$ when $Y \sim p$.

If given, we use $\ell : \mathcal{R} \times \mathcal{Y} \to \mathbb{R}$ to denote a *target loss*, with predictions $r \in \mathcal{R}$. Similarly, $L : \mathbb{R}^d \times \mathcal{Y} \to \mathbb{R}$ will typically denote a *surrogate* loss, with surrogate predictions $u \in \mathbb{R}^d$. In this case, $d$ is the prediction dimension of $d$. We write $\mathcal{L}_d$ for the set of (Borel) $\mathcal{B}(\mathbb{R}^d) \otimes \mathcal{Y}$-measurable and lower semi-continuous surrogates $L : \mathbb{R}^d \times \mathcal{Y} \to \mathbb{R}$ such that $\mathbb{E}_{Y \sim p} L(u, Y) < \infty$ for all $u \in \mathbb{R}^d, p \in \mathcal{P}$, that are minimizable in that $\arg\min_u \mathbb{E}_p L(u, Y)$ is nonempty for all $p \in \mathcal{P}$. (See § F.1 for a discussion of this assumption.) Moreover, $\mathcal{L}_d^{\text{cvx}} \subseteq \mathcal{L}_d$ is the set of convex (in $\mathbb{R}^d$ for every $y \in \mathcal{Y}$) losses in $\mathcal{L}_d$. Set $\mathcal{L} = \cup_{d \in \mathbb{N}} \mathcal{L}_d$, and $\mathcal{L}^{\text{cvx}} = \cup_{d \in \mathbb{N}} \mathcal{L}_d^{\text{cvx}}$. A loss $\ell : \mathcal{R} \times \mathcal{Y} \to \mathbb{R}$ is *discrete* if $\mathcal{R}$ is a finite set.

### 2.1   Property elicitation

Arising from the statistics and economics literature, property elicitation is similar to calibration, but only characterizes exact minimizers of a surrogate [17, 18, 25–27, 30, 37]. Specifically, given a statistic or *property* $\Gamma$ of interest, which maps a distribution $p \in \mathcal{P} \subseteq \Delta_\mathcal{Y}$ to the set of desired or correct predictions, the minimizers of $L$ should precisely coincide with $\Gamma$. For example, squared loss $L(r, y) = (r - y)^2$ elicits the mean $\Gamma(p) = \{\mathbb{E}_p Y\}$.

**Definition 1** (Property, elicits). *A property is a set-valued function* $\Gamma : \mathcal{P} \to 2^\mathcal{R} \setminus \{\emptyset\}$, *which we denote* $\Gamma : \mathcal{P} \rightrightarrows \mathcal{R}$. *A loss* $L : \mathcal{R} \times \mathcal{Y} \to \mathbb{R}$ *elicits the property* $\Gamma$ *if*

$$\forall p \in \mathcal{P}, \quad \Gamma(p) = \arg\min_{u \in \mathcal{R}} \mathbb{E}_p L(u, Y) \ . \tag{1}$$

The *level set* of $\Gamma$ at value $r \in \mathcal{R}$ is $\Gamma_r := \{p \in \mathcal{P} : r \in \Gamma(p)\}$. We call a property $\Gamma : \mathcal{P} \rightrightarrows \mathcal{R}$ *discrete* if $\mathcal{R}$ is a finite set, as in Quadrants 1 and 2. A property is *single-valued* if $|\Gamma(p)| = 1$ for all $p \in \mathcal{P}$, in which case we may write $\Gamma : \mathcal{P} \to \mathcal{R}$ and $\Gamma(p) \in \mathcal{R}$. As an example, the mean is single-valued. We define the *range* of a property by range $\Gamma = \bigcup_{p \in \mathcal{P}} \Gamma(p) \subseteq \mathcal{R}$. When $L \in \mathcal{L}$, we use $\Gamma := \text{prop}_\mathcal{P}[L]$ to denote the unique property elicited by $L$ (for distributions in $\mathcal{P}$) from eq. (1). Typically, we denote the target property by $\gamma$, and the surrogate by $\Gamma$.

## 2.2 Consistency and indirect elicitation

As discussed above, notions of consistency have appeared in the literature with respect to target losses, and to target statistics or properties. We give informal definitions of both notions here, with formal versions deferred to § A.

**Definition 2** (Consistent: loss (informal)). *A loss $L \in \mathcal{L}$ and link $(L, \psi)$ are* consistent *with respect to a target loss $\ell$ if, for all distributions $D$ over $\mathcal{X} \times \mathcal{Y}$ and all sequences of measurable hypothesis functions $\{f_m : \mathcal{X} \to \mathcal{R}\}$,*

$$\mathbb{E}_D L(f_m(X), Y) \to \inf_f \mathbb{E}_D L(f(X), Y) \implies \mathbb{E}_D \ell((\psi \circ f_m)(X), Y) \to \inf_f \mathbb{E}_D \ell((\psi \circ f)(X), Y) .$$

Consistency with respect to a property follows similarly, but instead of converging to the optimal target loss, one should approach the optimal (conditional) property value.

**Definition 3** (Consistent: property (informal)). *Suppose we are given a loss $L \in \mathcal{L}$, link function $\psi : \mathbb{R}^d \to \mathcal{R}$, and property $\gamma : \mathcal{P} \rightrightarrows \mathcal{R}$. Moreover, let $\mu : \mathcal{R} \times \mathcal{P} \to \mathbb{R}_+$ be any function satisfying $\mu(r, p) = 0 \iff r \in \gamma(p)$. We say $(L, \psi)$ is* consistent *with respect to $\gamma$ if, there exists a $\mu$ such that, for all $D$ over $\mathcal{X} \times \mathcal{Y}$ and sequences of measurable functions $\{f_m : \mathcal{X} \to \mathcal{R}\}$,*

$$\mathbb{E}_D L(f_m(X), Y) \to \inf_f \mathbb{E}_D L(f(X), Y) \implies \mathbb{E}_X \mu(\psi \circ f_m(X), D_X) \to 0 . \tag{2}$$

Lemma 1 in § A shows that, in fact, one can capture consistency with respect to a target loss as a special case of consistency with respect to a target property. Specifically, given a target loss $\ell$, one can take $\gamma = \mathrm{prop}_{\mathcal{P}}[\ell]$ and define $\mu(r, p) := \mathbb{E}_p \ell(r, Y) - \min_{r'} \mathbb{E}_p \ell(r', Y)$ to be the $\ell$-regret of the report $r$. This observation allows us to translate consistency from Quadrant 1 to Quadrant 2, and from Quadrant 3 to Quadrant 4; in particular, it will allow us to prove bounds for all four quadrants simultaneously.

As observed in the literature, e.g. [2, 40], both notions of consistency imply in particular that the link function must map exactly optimal surrogate reports to exactly optimal target reports. In property elicitation, this condition is known as *indirect elicitation*: for single-valued properties, $\Gamma$ and $\psi$ indirectly elicit $\gamma$ if $\gamma = \psi \circ \Gamma$. The definition below covers the general set-valued case as well.

**Definition 4** (Indirect Elicitation). *A surrogate loss and link $(L, \psi)$* indirectly elicit *a property $\gamma : \mathcal{P} \rightrightarrows \mathcal{R}$ if $L$ elicits a property $\Gamma : \mathcal{P} \rightrightarrows \mathbb{R}^d$ such that for all $u \in \mathbb{R}^d$, we have $\Gamma_u \subseteq \gamma_{\psi(u)}$. We say $L$ indirectly elicits $\gamma$ if such a link $\psi$ exists.*

**Proposition 1.** *For a surrogate $L \in \mathcal{L}$, if the pair $(L, \psi)$ is consistent with respect to a property $\gamma : \mathcal{P} \rightrightarrows \mathcal{R}$ or a loss $\ell$ eliciting $\gamma$, then $(L, \psi)$ indirectly elicits $\gamma$.*

In other words, indirect elicitation is a necessary condition for consistency. In light of Lemma 1, we can use this fact to build prediction dimension lower bounds across all four quadrants.

Implicit in the above elicitation definitions is that $L$ is minimizable: since $\Gamma = \mathrm{prop}_{\mathcal{P}}[L]$ is nonempty everywhere, the expected loss $\mathbb{E}_p L(\cdot, Y)$ always achieves a minimum. This restriction is also implicit in previous work, e.g., [2]. See § F.1 for further discussion.

## 2.3 Convex consistency dimension and elicitation complexity

Various works have studied the minimum prediction dimension $d$ needed in order to construct a consistent surrogate loss $L : \mathbb{R}^d \times \mathcal{Y} \to \mathbb{R}$, typically through proxies such as calibration [2, 33, 40] and property elicitation [15, 18, 20]. Motivated by the importance of convex surrogates in machine learning, Ramaswamy and Agarwal [33] introduce the following definition for Quadrant 1; we generalize it to all quadrants.

**Definition 5** (Convex Consistency Dimension). *Given target loss $\ell : \mathcal{R} \times \mathcal{Y} \to \mathbb{R}$ or property $\gamma : \mathcal{P} \rightrightarrows \mathcal{R}$, its* convex consistency dimension $\mathrm{cons}_{\mathrm{cvx}}(\cdot)$ *is the minimum dimension $d$ such that $\exists L \in \mathcal{L}_d^{\mathrm{cvx}}$ and link $\psi$ such that $(L, \psi)$ is consistent with respect to $\ell$ or $\gamma$.*

In the case of a target property $\gamma$, Lambert et al. [27] similarly introduce the notion of *elicitation complexity*. Later generalized by Frongillo and Kash [20], elicitation complexity is the lowest prediction dimension of an elicitable property, from some class of properties, from which one can compute $\gamma$. We give here the definition for convex-elicitable properties.

**Definition 6** (Convex Elicitation Complexity)**.** *Given a target property $\gamma$, the* convex elicitation complexity $\mathrm{elic}_{\mathrm{cvx}}(\gamma)$ *is the minimum dimension $d$ such that there is a $L \in \mathcal{L}_d^{\mathrm{cvx}}$ indirectly eliciting $\gamma$.*

As consistency implies indirect elicitation, we have the following.

**Corollary 1.** *Given a property $\gamma : \mathcal{P} \rightrightarrows \mathcal{R}$ or loss $\ell : \mathcal{R} \times \mathcal{Y} \to \mathbb{R}$ eliciting $\gamma$, we have* $\mathrm{elic}_{\mathrm{cvx}}(\gamma) \leq \mathrm{cons}_{\mathrm{cvx}}(\gamma) = \mathrm{cons}_{\mathrm{cvx}}(\ell)$.

Finally, related to our work is the *embedding dimension* of Finocchiaro et al. [12], which is a lower bound on both convex elicitation complexity of discrete properties and convex consistency dimension of discrete losses and finite statistics.

## 3 Lower bounding convex consistency dimension via $d$-flats

We now turn to the question of bounding the convex consistency dimension for a given task. From Proposition 1, given a target property $\gamma$ or loss $\ell$ with $\gamma = \mathrm{prop}_{\mathcal{P}}[\ell]$, this task reduces to lower bounding the convex consistency dimension of $\gamma$. Theorem 1, crystallized from the proofs of Ramaswamy and Agarwal [33, Theorem 16] and Agarwal and Agarwal [2, Theorem 9], considers a particular distribution $p$ and surrogate prediction $u \in \mathbb{R}^d$ which is optimal for $p$. Theorem 1 will show that if $d$ is small, then the level set $\{p \in \mathcal{P} : u \in \arg\min_{u'} \mathbb{E}_p L(u', Y)\}$ must be large; in fact, it must roughly contain a high-dimensional *flat* (of codimension $d$). By definition of indirect elicitation, there is some level set $\gamma_r$ (where $u$ is linked to $r$) containing this flat as well. We can then leverage the contrapositive of this result: if $\gamma$ has a level set intricate enough not to contain any high-dimensional flats, then $\gamma$ cannot have a low-dimensional consistent convex surrogate.

**Definition 7** ($d$-flat)**.** *For $d \in \mathbb{N}$, a $d$-flat, or simply* flat, *is a nonempty set $F = \ker_{\mathcal{P}} W := \{q \in \mathcal{P} : \mathbb{E}_q W = \vec{0}\}$ for some measurable $W : \mathcal{Y} \to \mathbb{R}^d$.*

The following lemma yields consistency bounds when combined with Proposition 1. A similar result is found in Agarwal and Agarwal [2, Theorem 9], which bounds the dimension of level sets of a single-valued $\mathrm{prop}_{\mathcal{P}}[L]$. Theorem 1 instead bounds the dimension of flats contained in the level sets, an additional power which we leverage in our examples.

**Theorem 1.** *Let $\Gamma : \mathcal{P} \rightrightarrows \mathbb{R}^d$ be (directly) elicited by $L \in \mathcal{L}_d^{\mathrm{cvx}}$ for some $d \in \mathbb{N}$. Let $\mathcal{Y}$ be either a finite set, or $\mathcal{Y} = \mathbb{R}$, in which case we assume each $p \in \mathcal{P}$ admits a Lebesgue density supported on the same set for all $p \in \mathcal{P}$.[1] For all $u \in \mathrm{range}\,\Gamma$ and $p \in \Gamma_u$, there is some $d$-flat $F$ such that $p \in F \subseteq \Gamma_u$.*

*Proof (finite case).* We will prove the result for the case of finite $\mathcal{Y}$, and defer the $\mathcal{Y} = \mathbb{R}$ case to § B. As $L$ is convex and elicits $\Gamma$, we have $u \in \Gamma(p) \iff \vec{0} \in \partial \mathbb{E}_p L(u, Y)$. With $\mathcal{Y}$ finite, this is additionally equivalent to $\vec{0} \in \oplus_y p_y \partial L(u, y)$, where $\oplus$ denotes the Minkowski sum [23, Theorem 4.1.1].[2] Expanding, we have $\oplus_y p_y \partial L(u, y) = \{\sum_{y \in \mathcal{Y}} p_y x_y \mid x_y \in \partial L(u, y)\ \forall y \in \mathcal{Y}\}$, and thus there is a $W$ such that $Wp = \sum_y p_y x_y = \vec{0}$ where $W = [x_1, \ldots, x_n] \in \mathbb{R}^{d \times n}$; cf. [33, $\mathbf{A}^m$ in Theorem 16]. Let $V_{u,p} : \mathcal{Y} \to \mathbb{R}^d, y \mapsto W_y$ be the function encoding the columns of $W$. Observe that $\mathbb{E}_p V_{u,p} = \vec{0}$. We take the flat $F := \ker_{\mathcal{P}} V_{u,p}$, and have $p \in F$ by construction. To see $F \subseteq \Gamma_u$, from the chain of equivalences above, we have for any $q \in \mathcal{P}$ that $q \in \ker_{\mathcal{P}} V_{u,p} \implies \vec{0} \in \partial \mathbb{E}_q L(u, Y) \implies u \in \Gamma(q) \implies q \in \Gamma_u$. $\square$

---

[1]This assumption is largely for technical convenience, to ensure that $\mathcal{V}_{u,p}$ does not depend on $p$. Any such assumption would suffice, and we suspect even that condition can be relaxed.

[2]$\partial$ represents the subdifferential $\partial f(x) = \{z : f(x') - f(x) \geq \langle z, x' - x \rangle\ \forall x'\}$.

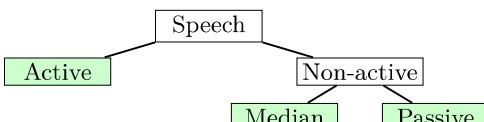

Figure 2: Hierarchical prediction example with labeling tone of speech. We take $\mathcal{Y} = \mathcal{R}$ to be the leaves of this tree, shown in blue.

Theorem 1 now allows us to derive bounds on convex consistency dimension by considering distributions and property values that are either single-valued (Corollary 2) or on the relative interior of the simplex with finite $\mathcal{Y}$ (Corollary 3). Proofs are deferred to § B.

**Corollary 2.** *Let target property $\gamma : \mathcal{P} \rightrightarrows \mathcal{R}$ and $d \in \mathbb{N}$ be given. Let $\mathcal{Y}$ be either a finite set, or $\mathcal{Y} = \mathbb{R}$, in which case we assume each $p \in \mathcal{P}$ admits a Lebesgue density supported on the same set for all $p \in \mathcal{P}$. Let $p \in \mathcal{P}$ with $|\gamma(p)| = 1$, and take $\gamma(p) = \{r\}$. If there is no $d$-flat $F$ with $p \in F \subseteq \gamma_r$, then $\mathrm{cons}_{\mathrm{cvx}}(\gamma) \geq \mathrm{elic}_{\mathrm{cvx}}(\gamma) \geq d+1$.*

**Corollary 3.** *Let an elicitable target property $\gamma : \mathcal{P} \rightrightarrows \mathcal{R}$ be given, where $\mathcal{P} \subseteq \Delta_{\mathcal{Y}}$ is defined over a finite set of outcomes $\mathcal{Y}$, and let $d \in \mathbb{N}$. Let $p \in \mathrm{relint}(\mathcal{P})$. If there is no $d$-flat $F$ with $p \in F \subseteq \gamma_r$, then $\mathrm{cons}_{\mathrm{cvx}}(\gamma) \geq \mathrm{elic}_{\mathrm{cvx}}(\gamma) \geq d+1$.*

### 3.1 Illustrating the condition in all four quadrants

We now illustrate how to apply Theorem 1 to construct lower bounds on convex consistency dimension for targets across all four quadrants of Table 1. Throughout the examples, we will have $|\mathcal{Y}| = 3$ so that the probability simplex can be visualized in two dimensions (Figure 3). For each, we take $d = 1$, and thus ask whether any 1-flat (a line in the figures) passes through the point $p$ while staying within the corresponding level set.

**Q1: Classification with an abstain option.** The abstain target loss is a well-studied variation of 0-1 loss that allows for an "abstain" report that gives a lesser punishment $1/2$ for abstaining, $r = \perp$ [7, 8, 29, 33, 34]. Formally, the target loss is $\ell^{1/2}(r, y) := \mathbf{I}\{r \notin \{y, \perp\}\} + (1/2)\mathbf{I}\{r = \perp\}$. Since we are given a discrete target loss, this problem fits nicely into Quadrant 1.

To apply Theorem 1, we first consider the abstain property $\gamma$ elicited by $\ell^{1/2}$, where one predicts the most likely outcome $y$ if $Pr[Y = y] \geq 1/2$ and otherwise "abstains" by predicting $\perp$. For the depicted distribution $p \in \mathrm{relint}(\gamma_\perp)$, we cannot fit a 1-flat (line) fully contained in $\gamma_\perp$ that passes through $p$. By Corollary 3, we can conclude $\mathrm{cons}_{\mathrm{cvx}}(\gamma^{1/2}) \geq 2$ when $|\mathcal{Y}| = 3$, meaning there is no consistent convex surrogate in 1 dimension. This lower bound matches the upper bound from the convex surrogate of Ramaswamy and Agarwal [33].

**Q2: Variation of hierarchical classification.** Ramaswamy et al. [32] study hierarchical classification tasks, in which labels are arranged in a tree and one wishes to predict the deepest node in a tree that is "likely enough" [5, 43]. Consider the variation of this task where one can only predict leaves of this tree. For example, Figure 2 depicts a speech classification task where speech is either active or non-active, and non-active is further subdivided into median and passive. It is natural to predict active if that label is more likely than both non-active labels combined, and otherwise to predict the most likely of median and passive:

$$\gamma(p) = \begin{cases} \text{active} & p_{\text{active}} \geq 1/2 \\ \text{median} & p_{\text{active}} \leq 1/2 \wedge p_{\text{median}} \geq p_{\text{passive}} \\ \text{passive} & p_{\text{active}} \leq 1/2 \wedge p_{\text{passive}} \geq p_{\text{median}} \end{cases} .$$

This "T-shaped" property, depicted in Figure 3 (Q2), falls under Quadrant 2, as it is not elicited by any target loss.[3] Like abstain, we cannot fit a 1-flat (line) entirely contained in the level set $\gamma_{\text{passive}}$ through the depicted $p$, so Corollary 3 gives $\mathrm{cons}_{\mathrm{cvx}}(\gamma) = 2$.

**Q3: Least-squares regression** Squared loss is commonly used in machine learning and statistics for continuous estimation, making it the canonical choice for Quadrant 3.

---

[3]The cells of finite elicitable properties form power diagrams, a generalization of Voronoi diagrams, which disallow this "T-shaped" configuration [17, 26].

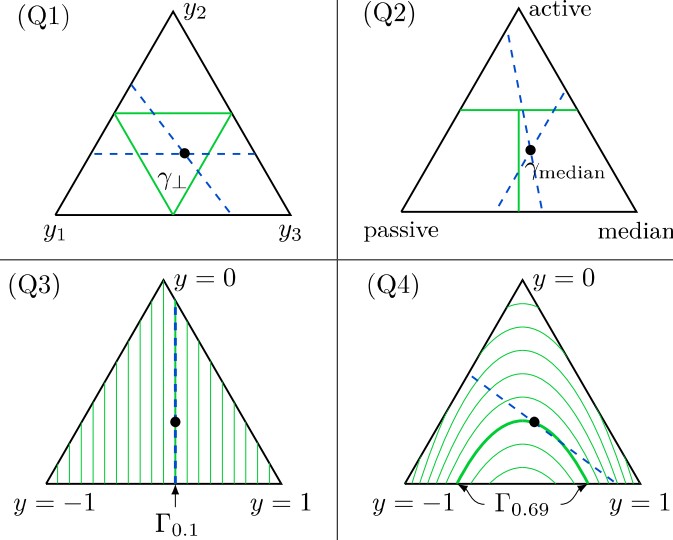

Figure 3: Example properties for each quadrant. Throughout, we take ● to be the distribution $p = (0.3, 0.3, 0.4)$ according to the left, top, and right outcomes respectively. (Q1,Q2) We cannot fit a 1-flat (line) through $p$ without leaving the level sets $\gamma_\perp$ and $\gamma_{\text{median}}$, respectively; Theorem 1 implies that there is no 1-dimensional consistent convex surrogate for either problem. (Q3) Squared error is a 1-dimensional convex loss, and indeed it elicits the mean of $Y$, whose level sets are all 1-flats. (Q4) The level sets of the variance are curved and cannot fit a 1-flat; from Theorem 1 there is no 1-dimensional convex surrogate consistent for the variance.

Squared loss is a 1-dimensional convex loss which elicits the mean $\Gamma(p) = \mathbb{E}_p[Y]$. Theorem 1 therefore states that we can fit a 1-flat through any distribution $p$ while staying within the corresponding level set. In fact, the level sets of the mean are all exactly 1-flats, as demonstrated in Figure 3 (Q3).

**Q4: Variance** Consider the task of estimating the variance $\text{Var}(p) = \mathbb{E}_p[Y^2] - \mathbb{E}_p[Y]^2$. The variance is not (directly) elicitable as its level sets are not convex [27, 31], meaning this task falls under Quadrant 4. Interestingly, the fact that the variance is not elicitable does not yield a lower bound on elicitation complexity of 2, as it does not rule out the variance being a link of a real-valued convex-elicitable property; cf. Frongillo and Kash [20, Remark 1]. In § F.2, we show $\text{elic}_{\text{cvx}}(\text{Var}) = 2$, meaning the lowest dimension of a convex loss to estimate conditional variance is 2. This lower bound will follow from Theorem 2 in § 4 using the fact that variance is the Bayes risk of squared loss. While perhaps intuitively obvious, even this simple result is novel.

## 3.2 Relation to feasible subspace dimension

In Quadrant 1, Ramaswamy and Agarwal [33] give a lower bound on convex consistency dimension roughly by the co-dimension of the *subspace of feasible directions* $\mathcal{S}_{\mathcal{C}}(p)$ of a convex set $\mathcal{C}$ at a given distribution $p$ such that $p \in \mathcal{C}$, which is loosely the "most full" subspace of $\mathcal{C}$ containing a neighborhood around $p$.

$$\mathcal{S}_{\mathcal{C}}(p) = \{v \in \mathbb{R}^n \mid \exists \epsilon_0 > 0 \text{ such that } p + \epsilon v \in \mathcal{C} \forall \epsilon \in (-\epsilon_0, \epsilon_0)\}$$

Theorem 1 subsumes the bounds given by Ramaswamy and Agarwal [33] by showing that, if there is a $d$-flat through $p$ fully contained in a level set $\gamma_r$ (so we can apply Theorem 1) then the subspace of feasible directions at the same $p \in \mathcal{C} := \gamma_r$ has co-dimension at most $d$, discussed in detail in § D.1.

**Proposition 2.** *Suppose we are given a discrete loss $\ell : \mathcal{R} \times \mathcal{Y} \to \mathbb{R}$ eliciting the property $\gamma : \Delta_{\mathcal{Y}} \rightrightarrows \mathcal{R}$. Fix $p \in \text{relint}(\Delta_{\mathcal{Y}})$ and take $r \in \mathcal{R}$ such that $p \in \gamma_r$. If $\text{cons}_{\text{cvx}}(\ell) = d$, then there exists a $d$-flat $F \subseteq \gamma_r$ through $p$. Moreover, $F$ is a subspace of feasible directions over the set $\gamma_r$ intersected with the simplex. Therefore, $\text{codim}(\mathcal{S}_{\gamma_r}(p)) \leq d$, and in turn, this implies $\text{cc dim}(\ell) \geq d \geq \text{codim}(\mathcal{S}_{\gamma_r}(p))$.*

In other words, any $d$-flat through $p$ is a subspace of feasible directions of co-dimension at most $d$, so Theorem 1 provides a weakly tighter lower bound on convex consistency dimension than Ramaswamy and Agarwal [33, Theorem 16]. In fact, the $d$-flats bound can be strictly tighter; in § D we show that the abstain example from Figure 3 (Q1) yields a $d$-flats lower

bound of 2 and a feasible subspace dimension lower bound of 1. This gap stems from the fact that feasible subspace dimension uses only local information of the property to construct lower bounds, while $d$-flats in Theorem 1 allow us to additionally use global information. See Figure 4 in § D for an illustration.

# 4 Application: Risk Measures, Mode, and Modal Interval

We now turn to two main applications of Theorem 1: new lower bounds on the convex consistency dimension of risk measures (§ 4.1) and the mode and modal interval (§ 4.2). In both cases, we build on previous results due to Frongillo and Kash [19, 20] and Dearborn and Frongillo [10] which showed lower bounds with respect to *identifiable* properties; a property is $d$-identifiable if its level sets are all $d$-flats, as in Figure 3 (Q3). In contrast, properties elicited by convex losses are generally not identifiable, particularly when the loss is non-smooth. For example, the properties elicited by hinge loss and the abstain surrogate are not identifiable, as their level sets are not flats; see Figure 3 (Q1). It therefore might appear that entirely new ideas are needed. Indeed, both papers above pose developing similar bounds with respect to convex-elicitable properties as a major open question.

Using our $d$-flats framework, we resolve both open questions with new lower bounds in both settings. Our framework clarifies the relationship between $d$-identifiable properties and properties elicited by $d$-dimensional convex losses: the level sets of the former are $d$-flats by definition, while the level sets of the latter are *unions* of $d$-flats by Theorem 1. A careful examination of the arguments of Frongillo and Kash [19, 20] and Dearborn and Frongillo [10] reveals that they largely rely on the containment of $d$-flats in level sets, rather than the full structure of identifiable properties. As such, although quite subtle in the case of risk measures, the general structure of these previous proofs go through for convex-elicitable properties: since no $d$-flat could be contained in a particular level set, no union of $d$-flats could be either. Our lower bounds therefore match both of these papers, though we conjecture that our convex consistency bounds could be tightened in some cases.

## 4.1 Risk measures (Q4)

The problem of estimating a risk or uncertainty measure of $Y$ is of central importance in financial regulation [1, 6, 14] and robust engineering design [4, 35, 38]. Risk measures include the upper confidence bound $\mathbb{E}[Y] + \lambda\sqrt{\mathrm{Var}[Y]}$, or the conditional value at risk (CVaR) defined below in eq. (3), in either conditional or unconditional contexts. Uncertainty measures include the variance, entropy, or norm of the distribution of $Y$. Risk and uncertainty measures are typically not elicitable, so this problem falls under Quadrant 4. Frongillo and Kash [19, 20] give prediction dimension lower bounds for a broad class of risk and uncertainty measures, namely Bayes risks. As stated above, these bounds are with respect to identifiable properties, and bounds for convex surrogates are left as a major open question.

We resolve this open question using our $d$-flats framework, giving a matching result for convex-elicitable properties (Theorem 2). First we recall the definition of the Bayes risk.

**Definition 8.** *Given loss function* $L : \mathcal{R} \times \mathcal{Y} \to \mathbb{R}$ *for some report set* $\mathcal{R}$*, the* Bayes risk *of* $L$ *is defined as* $\underline{L}(p) := \inf_{r \in \mathcal{R}} \mathbb{E}_p L(r, Y)$.

**Condition 1.** *For some* $r \in \mathrm{range}\,\Gamma$*, the level set* $\Gamma_r = \ker_\mathcal{P} V$ *is a* $d$*-flat presented by some* $V : \mathcal{Y} \to \mathbb{R}^d$ *such that* $0 \in \mathrm{int}\,\{\mathbb{E}_p V : p \in \mathcal{P}\}$.

**Theorem 2.** *Let* $\mathcal{P}$ *be a convex set of Lebesgue densities supported on the same set for all* $p \in \mathcal{P}$*. Let* $\Gamma : \mathcal{P} \to \mathbb{R}^d$ *satisfy Condition 1 for some* $r \in \mathbb{R}^d$*. Let* $L \in \mathcal{L}^{\mathrm{cvx}}$ *elicit* $\Gamma$ *such that* $\underline{L}$ *is non-constant on* $\Gamma_r$*. Then* $\mathrm{cons}_{\mathrm{cvx}}(\underline{L}) \geq \mathrm{elic}_{\mathrm{cvx}}(\underline{L}) \geq d + 1$.

To illustrate the theorem, we briefly apply it to one of the most prominent financial risk measures, the conditional value at risk (CVaR). Several other applications from Frongillo and Kash [19, 20], such as other risk measures, entropy, and norms, follow similarly. The authors observe that CVaR can be expressed as a Bayes risk; for $0 < \alpha < 1$, we may define

$$\mathrm{CVaR}_\alpha(p) = \inf_{r \in \mathbb{R}} \mathbb{E}_p \left\{ \tfrac{1}{\alpha}(r - Y)\mathbb{1}_{r \geq Y} - r \right\} , \tag{3}$$

which is the Bayes risk of the transformed pinball loss $L_\alpha(r, y) = \frac{1}{\alpha}(r - y)\mathbb{1}_{r \geq y} - r$. In turn, $L_\alpha$ elicits the $\alpha$-quantile, the quantity $q_\alpha(p)$ such that $\Pr_p[Y \geq q_\alpha(p)] = \alpha$. Following Frongillo and Kash [20], we will restrict to the set $\mathcal{P}_q$ of probability measures over $\mathbb{R}$ with connected support and whose CDFs are strictly increasing on their support, so that $q_\alpha$ is single-valued. Under mild assumptions, we find that there is no consistent real-valued convex surrogate for $\mathrm{CVaR}_\alpha$.

**Corollary 4.** *Let $\mathcal{P}$ be a convex set of continuous Lebesgue densities on $\mathcal{Y} = \mathbb{R}$ with all $p \in \mathcal{P}$ having support on the same interval. If we have $p_1, p_2, p_3, p_2' \in \mathcal{P}$ with $q_\alpha(p_1) < q_\alpha(p_2) < q_\alpha(p_3)$ and $\mathrm{CVaR}_\alpha(p_2) \neq \mathrm{CVaR}_\alpha(p_2')$, then $\mathrm{cons}_{\mathrm{cvx}}(\mathrm{CVaR}_\alpha) \geq \mathrm{elic}_{\mathrm{cvx}}(\mathrm{CVaR}_\alpha) \geq 2$.*

As first shown by Fissler et al. [15], the pair $(\mathrm{CVaR}_\alpha, q_\alpha)$ is jointly identifiable and elicitable, but not by any convex loss [13, Prop. 4.2.31]. We conjecture the stronger statement $\mathrm{elic}_{\mathrm{cvx}}(\mathrm{CVaR}_\alpha) \geq 3$, which if true would constitute an interesting gap between elicitation complexity for identifiable and convex-elicitable properties.

## 4.2 Mode and modal interval (Q4, Q3)

For finite $|\mathcal{Y}|$, the mode $\gamma_{\mathrm{mode}}(p) = \arg\max_{y \in \mathcal{Y}} p(y)$ is elicited by 0-1 loss. By contrast, for $\mathcal{Y} = \mathbb{R}$, the mode is not elicitable [22], landing it in Quadrant 4. Defining the mode is subtle for general distributions; here let us assume $p$ has a smooth and bounded Lesbegue density $f_p$, and define the mode the same way, $\gamma_{\mathrm{mode}}(p) = \arg\max_{y \in \mathcal{Y}} f_p(y)$. Dearborn and Frongillo [10] recently showed a strong impossibility result, that the mode has countably infinite elicitation complexity with respect to identifiable properties. In other words, it is as hard to elicit the mode as the full distribution $p$ itself. Complexity with respect to convex-elicitable properties is left as an important open question.

We resolve this question, with a matching infinite lower bound for convex-elicitable properties. In light of our $d$-flats framework, the result is nearly immediate, as the proof in Dearborn and Frongillo [10] already showed that the level sets of the mode cannot contain any $d$-flats.

**Theorem 3.** *The mode has $\mathrm{cons}_{\mathrm{cvx}}(\gamma_{\mathrm{mode}}) = \mathrm{elic}_{\mathrm{cvx}}(\gamma_{\mathrm{mode}}) = \infty$ (countably infinite) with respect to $\mathcal{P}$, the class of probability measures on $\mathcal{Y} = \mathbb{R}$ with a smooth and bounded density and such that $\gamma_{\mathrm{mode}}$ is single-valued.*

*Proof.* The proof of Dearborn and Frongillo [10, Theorem 1] gives a distribution $p \in \mathcal{P}$ with $\gamma_{\mathrm{mode}}(p) = 0 =: u$. It then introduces an arbitrary identification function $V : \hat{\mathcal{R}} \times \mathcal{Y} \to \mathbb{R}^k$, $k \in \mathbb{N}$, and value $r \in \hat{\mathcal{R}}$ such that $p \in \ker_{\mathcal{P}} V(r, \cdot)$. Letting $F = \ker_{\mathcal{P}} V(r, \cdot)$, we therefore have an arbitrary $k$-flat containing $p$. The proof then proceeds to construct some $p' \in F$ with $\gamma_{\mathrm{mode}}(p') \neq u$. Corollary 3 now gives $\mathrm{cons}_{\mathrm{cvx}}(\gamma_{\mathrm{mode}}) \geq \mathrm{elic}_{\mathrm{cvx}}(\gamma_{\mathrm{mode}}) \geq k + 1$. As $k$ was arbitrary, the result follows. □

A closely related property for any $\beta > 0$ is the (midpoint of the) modal interval of width $2\beta$, given by $\gamma_\beta(p) = \arg\max_{x \in \mathbb{R}} p([x - \beta, x + \beta])$. Interestingly, unlike the mode for $\mathcal{Y} = \mathbb{R}$, the modal interval is elicitable, by the target loss $\ell_\beta(r, y) = \mathbb{1}\{|r - y| > \beta\}$. The problem of estimating the modal interval therefore could be thought of as falling under Quadrant 3.

As observed in Dearborn and Frongillo [10, Corollary 1], the properties $\gamma_{\mathrm{mode}}$ and $\gamma_\beta$ coincide with the family of distributions needed in their Theorem 1, meaning the conclusion of Theorem 3 transfers to the modal interval as well.

**Corollary 5.** *For any $\beta > 0$, the modal interval $\gamma_\beta : \mathcal{P}_\beta \to \mathbb{R}$ has $\mathrm{cons}_{\mathrm{cvx}}(\gamma_\beta) = \mathrm{elic}_{\mathrm{cvx}}(\gamma_\beta) = \infty$ (countably infinite) with respect to $\mathcal{P}_\beta$, the class probability measures on $\mathcal{Y} = \mathbb{R}$ with a smooth and bounded density, and such that $\gamma_{\mathrm{mode}}$ and $\gamma_\beta$ are single-valued.*

Thus, while $\gamma_\beta$ is elicitable, it does not have any consistent finite-dimensional convex surrogate. While this statement may seem counter-intuitive, recall that the mode for finite $|\mathcal{Y}|$ has $\mathrm{cons}_{\mathrm{cvx}}(\gamma_{\mathrm{mode}}) = |\mathcal{Y}| - 1$. Taking the limit as $|\mathcal{Y}| \to \infty$, one may therefore expect an infinite convex consistency dimension for both the mode and modal interval.

# 5 Conclusions and future work

In this work, we introduce a new tool to generate lower bounds on the convex consistency dimension of general prediction tasks. This tool is simultaneously broader, stronger, and easier to understand than previous results. Its breadth is demonstrated by applying to multiple problem types simultaneously (§ 3), while its strength is demonstrated by proving new bounds on convex consistency dimension (§ 4), and ease is apparent when observing that indirect elicitation is a strictly weaker notion than calibration – the most common proxy for consistency. We then apply our framework to yield new bounds on convex consistency dimension for entropy, risk measures, the mode, and modal intervals.

Several important questions remain open. Particularly for the discrete settings, we would like to know whether one can lift the restriction that surrogates always achieve a minimum; we conjecture positively (see § F.1). The observation that our bounds are as tight as calibration-based bounds, yet we use the weaker condition of indirect elicitation, motivates the study of how much weaker indirect elicitation is than calibration. More broadly, we would like to characterize $\text{cons}_{\text{cvx}}$ and $\text{elic}_{\text{cvx}}$ and develop a general framework for constructing surrogates achieving the best possible prediction dimension.

## Acknowledgments and Disclosure of Funding

We thank Adam Bloniarz and Nishant Mehta for several key suggestions and insights. This material is based upon work supported by the National Science Foundation under Grant No. IIS-2045347 and Graduate Research Fellowship No. DGE-1650115.

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
