# A  Consistency Implies Indirect Elicitation

In this section, we connect consistency of any surrogate to an indirect elicitation requirement. This will allow us to show indirect elicitation gives state-of-the-art lower bounds on convex consistency dimension.

We start by formalizing consistency in two ways that generalize across our four quadrants. First, given a target loss $\ell$, we say $L$ is consistent if optimizing $L$ and applying a link $\psi$ optimizes $\ell$ (Definition 9). Second, given a target property $\gamma$, such as the $\alpha$-quantile, we say $L$ is consistent if optimizing $L$ implies approaching, in some sense, the correct statistic $\gamma(D_x)$ of the conditional distributions $D_x = \Pr[Y|X = x]$ (Definition 10). We then observe that Definition 9 is subsumed by Definition 10, and use this to show consistency implies $L$ indirectly elicits $\mathrm{prop}_{\mathcal{P}}[\ell]$ or $\gamma$ respectively.

**Condition 2** (Covers). *A set $\mathcal{D} \subseteq \Delta(\mathcal{X} \times \mathcal{Y})$ covers a convex set $\mathcal{P} \subseteq \Delta_{\mathcal{Y}}$ if, for all $p \in \mathcal{P}$, there exists $D \in \mathcal{D}$ and $x \in \mathcal{X}$ such that $D$ has a point mass on $x$ and $p = D_x$.*

**Definition 9** (Consistent: loss). *A loss $L \in \mathcal{L}$ and link $(L, \psi)$ are $\mathcal{D}$-consistent for a set $\mathcal{D}$ of distributions over $\mathcal{X} \times \mathcal{Y}$ with respect to a target loss $\ell$ if, for all $D \in \mathcal{D}$ and all sequences of measurable hypothesis functions $\{f_m : \mathcal{X} \to \mathcal{R}\}$,*

$$\mathbb{E}_D L(f_m(X), Y) \to \inf_f \mathbb{E}_D L(f(X), Y) \implies \mathbb{E}_D \ell((\psi \circ f_m)(X), Y) \to \inf_f \mathbb{E}_D \ell((\psi \circ f)(X), Y) \ .$$

*For a given convex set $\mathcal{P} \subseteq \Delta_{\mathcal{Y}}$, we simply say $(L, \psi)$ is* consistent *if it is $\mathcal{D}$-consistent for some $\mathcal{D}$ covering $\mathcal{P}$.*

Instead of a target loss $\ell$, one may want to learn a target property, i.e. a conditional statistic such as the expected value, variance, or entropy. In this case, following the tradition in the statistics literature on conditional estimation [11, 21, 36], we formalize consistency as converging to the correct conditional estimates of the property. Convergence is measured by functions $\mu(r, p)$ that formalize how close $r$ is to "correct" for conditional distribution $p$. In particular we should have $\mu(r, p) = 0 \iff r \in \gamma(p)$.

**Definition 10** (Consistent: property). *Suppose we are given a loss $L \in \mathcal{L}$, link function $\psi : \mathbb{R}^d \to \mathcal{R}$, and property $\gamma : \mathcal{P} \rightrightarrows \mathcal{R}$. Moreover, let $\mu : \mathcal{R} \times \mathcal{P} \to \mathbb{R}_+$ be any function satisfying $\mu(r, p) = 0 \iff r \in \gamma(p)$. We say $(L, \psi)$ is $(\mu, \mathcal{D})$-consistent with respect to $\gamma$ if, for all $D \in \mathcal{D}$ and sequences of measurable functions $\{f_m : \mathcal{X} \to \mathcal{R}\}$,*

$$\mathbb{E}_D L(f_m(X), Y) \to \inf_f \mathbb{E}_D L(f(X), Y) \implies \mathbb{E}_X \mu(\psi \circ f_m(X), D_X) \to 0 \ . \tag{4}$$

*We simply say $(L, \psi)$ is $\mu$-consistent if it is $(\mu, \mathcal{D})$-consistent for some $\mathcal{D}$ covering $\mathcal{P}$. Additionally, we say $(L, \psi)$ is* consistent *if there is a $\mu$ such that $(L, \psi)$ is $\mu$-consistent.*

Typical definitions of consistency require $\mathcal{D}$ to be the set of all distributions over $\mathcal{X} \times \mathcal{Y}$, while our conditions are much weaker. As the main focus of this paper is lower bounds on the prediction dimension, i.e., showing that surrogates of a certain prediction dimension cannot exist, these weaker conditions translate to stronger impossibility statements.

Given a target loss $\ell$, we can define a statistic $\gamma$, the property it elicits. Intuitively, consistency of a surrogate $L$ with respect to $\ell$ and $\gamma$ are equivalent, i.e. in both cases estimates should converge to values that minimize $\ell$-loss. We formalize this by letting $\mu$ be the $\ell$-regret, $R_\ell := \mathbb{E}_p \ell(r, Y) - \min_{r'} \mathbb{E}_p \ell(r', Y)$, yielding Lemma 1.

**Lemma 1.** *Let a convex $\mathcal{P} \subseteq \Delta_{\mathcal{Y}}$ be given. Given a surrogate loss $L \in \mathcal{L}$, link $\psi$, and target loss $\ell$, set $\mu(r, p) := \mathbb{E}_p \ell(r, Y) - \min_{r'} \mathbb{E}_p \ell(r', Y)$ as the excess risk of $\ell$, $R_\ell$. Then there is a $\mathcal{D}$ covering $\mathcal{P}$ such that $(L, \psi)$ is $\mathcal{D}$-consistent with respect to $\ell$ if and only if $(L, \psi)$ is $(\mu, \mathcal{D})$-consistent with respect to $\gamma := \mathrm{prop}_{\mathcal{P}}[\ell]$.*

*Proof.* First, observe that $\mu(r, p) = 0 \iff \mathbb{E}_p \ell(r, Y) = \inf_{r' \in \mathcal{R}} \mathbb{E}_p \ell(r', Y) \iff r \in \gamma(p)$. Now suppose $(L, \psi)$ are consistent with respect to $\ell$, and take any sequence $\{f_m\}$ of measurable hypotheses. Rewriting the right-hand side of Definition 9,

$$\mathbb{E}_D \ell(\psi \circ f_m(X), Y) \to \inf_f \mathbb{E}_D \ell(\psi \circ f(X), Y) \tag{5}$$

$$\iff \mathbb{E}_X R_\ell(\psi \circ f_m(X), D_X) \to 0$$

$$\iff \mathbb{E}_X \mu(\psi \circ f_m(X), D_X) \to 0 \ . \tag{6}$$

Therefore, $\mathbb{E}_D L(f_m(X), Y) \to \inf_f \mathbb{E}_D L(f(X), Y)$ implies (5) if and only if it implies (6). Observe that the assumptions on $\mathcal{L}$ allow us to apply the Fubini-Tonelli Theorem [16, Theorem 2.37], which yields the equivalence of eq. 5 to the next line. $\qquad\square$

Because each target loss in $\mathcal{L}$ elicits some property, but not all target properties can be elicited by a loss (e.g. the variance), consistency with respect to a property is the strictly broader notion. In a loose sense, Proposition 1 lets us translate problems about target losses to be about the properties these losses elicit. This points to indirect elicitation as a natural necessary condition for consistency, as formalized in Proposition 1.

**Proposition 1.** *For a surrogate $L \in \mathcal{L}$, if the pair $(L, \psi)$ is consistent with respect to a property $\gamma : \mathcal{P} \rightrightarrows \mathcal{R}$ or a loss $\ell$ eliciting $\gamma$, then $(L, \psi)$ indirectly elicits $\gamma$.*

*Proof.* By Lemma 1, it suffices to show the result for consistency with respect to a property $\gamma$, setting $\gamma := \mathrm{prop}_{\mathcal{P}}[\ell]$ if $\ell$ is given instead. We show the contrapositive; suppose $(L, \psi)$ does not indirectly elicit $\gamma$, meaning we have some $p \in \mathcal{P}$ so that $u \in \Gamma(p)$ but $\psi(u) \notin \gamma(p)$, where $\Gamma := \mathrm{prop}_{\mathcal{P}}[L]$. Observe that we use the fact $\Gamma(p) \neq \emptyset$. By definition, if we had consistency, there must be some distribution $D$ on $\mathcal{X} \times \mathcal{Y}$ with a point mass on some $x \in \mathcal{X}$ and $D_x = p$. Consider a constant sequence $\{f_m\}$ with $f_m = f'$ such that $f'(x) = u$, so that $\mathbb{E}_D L(f_m(X), Y) = \mathbb{E}_{D_x} L(f_m(x), Y) = \mathbb{E}_p L(u, Y)$. Since $u \in \Gamma(p)$, we have $\mathbb{E}_p L(u, Y) = \inf_f \mathbb{E}_{D_x} L(f(x), Y) = \inf_f \mathbb{E}_D L(f(X), Y)$. In particular, we have $\mathbb{E}_D L(f_m(X), Y) \to \inf_f \mathbb{E}_D L(f(X), Y)$. However, we have $\mathbb{E}_X \mu(\psi \circ f_m(X), D_X) = \mu(f_m(x), p) = \mu(\psi(u), p) \neq 0$, since $\psi(u) \notin \gamma(p)$. Therefore $(L, \psi)$ is not consistent with respect to $\gamma$ (Definition 10). $\qquad\square$

This result allows us to state elicitation complexity as a lower bound for convex consistency dimension.

**Corollary 1.** *Given a property $\gamma : \mathcal{P} \rightrightarrows \mathcal{R}$ or loss $\ell : \mathcal{R} \times \mathcal{Y} \to \mathbb{R}$ eliciting $\gamma$, we have $\mathrm{elic}_{\mathrm{cvx}}(\gamma) \leq \mathrm{cons}_{\mathrm{cvx}}(\gamma) = \mathrm{cons}_{\mathrm{cvx}}(\ell)$.*

# B  Implications of Convex Indirect Elicitation Bounds

We start by fully proving Theorem 1.

**Theorem 1.** *Let $\Gamma : \mathcal{P} \rightrightarrows \mathbb{R}^d$ be (directly) elicited by $L \in \mathcal{L}_d^{\mathrm{cvx}}$ for some $d \in \mathbb{N}$. Let $\mathcal{Y}$ be either a finite set, or $\mathcal{Y} = \mathbb{R}$, in which case we assume each $p \in \mathcal{P}$ admits a Lebesgue density supported on the same set for all $p \in \mathcal{P}$.[4] For all $u \in \mathrm{range}\,\Gamma$ and $p \in \Gamma_u$, there is some $d$-flat $F$ such that $p \in F \subseteq \Gamma_u$.*

*Proof.* As $L$ is convex and elicits $\Gamma$, we have $u \in \Gamma(p) \iff \vec{0} \in \partial \mathbb{E}_p L(u, Y)$. We proceed in two cases, depending on $|\mathcal{Y}|$.

*Finite $\mathcal{Y}$:* If $\mathcal{Y}$ is finite, this is additionally equivalent to $\vec{0} \in \oplus_y p_y \partial L(u, y)$, where $\oplus$ denotes the Minkowski sum [23, Theorem 4.1.1].[5] Expanding, we have $\oplus_y p_y \partial L(u, y) = \{\sum_{y \in \mathcal{Y}} p_y x_y \mid x_y \in \partial L(u, y) \; \forall y \in \mathcal{Y}\}$, and thus $Wp = \sum_y p_y x_y = \vec{0}$ where $W = [x_1, \ldots, x_n] \in \mathbb{R}^{d \times n}$; cf. [33, $\mathbf{A}^m$ in Theorem 16]. Let $V_{u,p} : \mathcal{Y} \to \mathbb{R}^d, y \mapsto W_y$ be the function encoding the columns of $W$. Observe that $\mathbb{E}_p V_{u,p} = \vec{0}$.

*$\mathcal{Y} = \mathbb{R}$:* Any $L \in \mathcal{L}_d^{\mathrm{cvx}}$ satisfies the assumptions of [24], so we may interchange subdifferentiation and expectation. Specifically, letting $\mathcal{V}_{u,p} = \{V : \mathcal{Y} \to \mathbb{R}^d \mid V \text{ measurable}, V(y) \in \partial L(u, y) \; p\text{-a.s.}\}$, we have $\partial \mathbb{E}_p L(u, Y) = \{\int V(y) dp(y) \mid V \in \mathcal{V}_{u,p}\}$. As $\vec{0} \in \partial \mathbb{E}_p L(u, Y)$, in particular, there is some $V_{u,p} \in \mathcal{V}_{u,p}$ such that $\mathbb{E}_p V_{u,p} = 0$. For any $q \in \mathcal{P}$, as by assumption $q$ is supported on the same set as $p$, we have $V_{u,p}(y) \in \partial L(u, y)$ $q$-a.s., so that $V_{u,p} \in \mathcal{V}_{u,q}$. Thus, $\mathbb{E}_q V_{u,p} = 0$ implies $0 \in \partial \mathbb{E}_q L(u, Y)$ by the above.

---

[4]This assumption is largely for technical convenience, to ensure that $\mathcal{V}_{u,p}$ does not depend on $p$. Any such assumption would suffice, and we suspect even that condition can be relaxed.

[5]$\partial$ represents the subdifferential $\partial f(x) = \{z : f(x') - f(x) \geq \langle z, x' - x \rangle \; \forall x'\}$.

In both cases, we take the flat $F := \ker_{\mathcal{P}} V_{u,p}$, and have $p \in F$ by construction. To see $F \subseteq \Gamma_u$, from the chain of equivalences above, we have for any $q \in \mathcal{P}$ that $q \in \ker_{\mathcal{P}} V_{u,p} \implies \vec{0} \in \partial \mathbb{E}_q L(u, Y) \implies u \in \Gamma(q) \implies q \in \Gamma_u$. $\qquad\square$

In order to apply Theorem 1 to various properties, we need the following lemmas about separating hyperplanes.

A hyperplane weakly separates two sets if its two closed halfspaces respectively contain the two sets.

**Lemma 2.** *If $\gamma : \mathcal{P} \rightrightarrows \mathcal{R}$ is an elicitable property, then for any pair of predictions $r, r' \in \mathcal{R}$ where $\gamma_r \neq \gamma_{r'}$, there is a hyperplane $H = \{x \in \mathbb{R}^{\mathcal{Y}} : v \cdot x = 0\}$, for some $v \in \mathbb{R}^{\mathcal{Y}}$, that weakly separates $\gamma_r$ and $\gamma_{r'}$ and has $\gamma_r \cap H = \gamma_{r'} \cap H = \gamma_r \cap \gamma_{r'}$.*

*Proof.* Let $\ell$ elicit $\gamma$. Let $v = \ell(r, \cdot) - \ell(r', \cdot)$, interpreted as a nonzero vector in $\mathbb{R}^{\mathcal{Y}}$. Let $H = \{q : v \cdot q = 0\}$. If $v \cdot q < 0$, then $r'$ cannot be optimal, so $q \notin \gamma_{r'}$. So $\gamma_{r'} \subseteq \{q : v \cdot q \geq 0\}$. Symmetrically, $\gamma_r \subseteq \{q : v \cdot q \leq 0\}$. This is weak separation, and it immediately implies that $\gamma_r \cap \gamma_{r'} \subseteq H$. Finally, if and only if $v \cdot q = 0$, i.e. $q \in H$, by definition the expected losses of both reports are the same. So $q \in \gamma_r \cap H \iff q \in \gamma_{r'} \cap H$. This gives $\gamma_r \cap H = \gamma_{r'} \cap H = \gamma_r \cap \gamma_{r'} \cap H = \gamma_r \cap \gamma_{r'}$. $\qquad\square$

**Lemma 3.** *Suppose we are given an elicitable property $\gamma : \mathcal{P} \rightrightarrows \mathcal{R}$, where $\mathcal{Y}$ is finite, and distribution $p \in \mathrm{relint}(\mathcal{P})$ such that $p \in \gamma_r \cap \gamma_{r'}$ for $r, r' \in \mathcal{R}$. Then for any flat $F$ containing $p$, $F \subseteq \gamma_r \iff F \subseteq \gamma_{r'}$.*

*Proof.* If $\gamma_r = \gamma_{r'}$, we are done. Otherwise, Lemma 2 gives a hyperplane $H = \{x \in \mathbb{R}^{\mathcal{Y}} : v \cdot x = 0\}$ and a guarantee that $\gamma_r \subseteq \{q \in \Delta_{\mathcal{Y}} : v \cdot q \leq 0\}$, while $\gamma_{r'} \subseteq \{q \in \Delta_{\mathcal{Y}} : v \cdot q \geq 0\}$, and finally $\gamma_r \cap \gamma_{r'} \subseteq H$.

Suppose $F \subseteq \gamma_r$; we wish to show $F \subseteq \gamma_{r'}$. Let $q \in F$. By Lemma 7(i), we have $p \in \mathrm{relint}(F)$, so there exists $\epsilon > 0$ so that $q' = p - \epsilon(q - p) \in F$.

Now, suppose for contradiction that $q \notin \gamma_{r'}$. Then $v \cdot q < 0$: containment in $\gamma_r$ gives $v \cdot q \leq 0$, and if $v \cdot q = 0$ then $q \in \gamma_r \cap H \implies q \in \gamma_{r'}$, a contradiction. But, noting that $p \in H$, we have $v \cdot q' = -\epsilon(v \cdot q) > 0$, so $q'$ is not in $\gamma_r$. This contradicts the assumption $F \subseteq \gamma_r$. Therefore, we must have $q \in \gamma_{r'}$, so we have shown $F \subseteq \gamma_{r'}$. Because $r$ and $r'$ were completely symmetric, this completes the proof. $\qquad\square$

Now we can understand the application of Theorem 1.

**Corollary 2.** *Let target property $\gamma : \mathcal{P} \rightrightarrows \mathcal{R}$ and $d \in \mathbb{N}$ be given. Let $\mathcal{Y}$ be either a finite set, or $\mathcal{Y} = \mathbb{R}$, in which case we assume each $p \in \mathcal{P}$ admits a Lebesque density supported on the same set for all $p \in \mathcal{P}$. Let $p \in \mathcal{P}$ with $|\gamma(p)| = 1$, and take $\gamma(p) = \{r\}$. If there is no $d$-flat $F$ with $p \in F \subseteq \gamma_r$, then $\mathrm{cons}_{\mathrm{cvx}}(\gamma) \geq \mathrm{elic}_{\mathrm{cvx}}(\gamma) \geq d + 1$.*

*Proof.* Let $(L, \psi)$ indirectly elicit $\gamma$, where $L \in \mathcal{L}_d^{\mathrm{cvx}}$, and let $\Gamma = \mathrm{prop}_{\mathcal{P}}[L]$. As $\Gamma$ is non-empty, there is some $u \in \Gamma(p)$. Since $\gamma$ is single-valued at $p$, we have $r = \psi(u)$; by Theorem 1, we know there is a $d$-flat $F = \ker_{\mathcal{P}} V_{u,p}$ so that $p \in F \subseteq \Gamma_u$. By definition of indirect elicitation, we additionally have $\Gamma_u \subseteq \gamma_r$. Thus, we have $p \in F \subseteq \gamma_r$. If no flat $F$ satisfies the above conditions, then no $L \in \mathcal{L}_d^{\mathrm{cvx}}$ indirectly elicits $\gamma$, so $\mathrm{elic}_{\mathrm{cvx}}(\gamma) \geq d + 1$, and recall $\mathrm{cons}_{\mathrm{cvx}}(\gamma) \geq \mathrm{elic}_{\mathrm{cvx}}(\gamma)$ by Corollary 1. $\qquad\square$

**Corollary 3.** *Let an elicitable target property $\gamma : \mathcal{P} \rightrightarrows \mathcal{R}$ be given, where $\mathcal{P} \subseteq \Delta_{\mathcal{Y}}$ is defined over a finite set of outcomes $\mathcal{Y}$, and let $d \in \mathbb{N}$. Let $p \in \mathrm{relint}(\mathcal{P})$. If there is no $d$-flat $F$ with $p \in F \subseteq \gamma_r$, then $\mathrm{cons}_{\mathrm{cvx}}(\gamma) \geq \mathrm{elic}_{\mathrm{cvx}}(\gamma) \geq d + 1$.*

*Proof.* Let $(L, \psi)$ indirectly elicit $\gamma$ and the convex function $L$ and elicit $\Gamma$. As $\Gamma$ is non-empty, there is some $u \in \Gamma(p)$, and suppose $r' = \psi(u)$. Take $F \subseteq \Gamma_u$ to be the flat that exists by Theorem 1. If $r = r'$, then $p \in F \subseteq \Gamma_u \subseteq \gamma_r$ by indirect elicitation. Otherwise, by Lemma 3, for elicitable properties with $p \in \gamma_r \cap \gamma_{r'}$, we observe $p \in F \subseteq \gamma_r \iff p \in F \subseteq \gamma_{r'}$.

As above, if no flat $F$ satisfies the above conditions, then no $L \in \mathcal{L}_d^{\mathrm{cvx}}$ indirectly elicits $\gamma$, so $\mathrm{cons}_{\mathrm{cvx}}(\gamma) \geq \mathrm{elic}_{\mathrm{cvx}}(\gamma) \geq d + 1$, recalling Corollary 1 for the first inequality. $\qquad\square$

## C Definitions of Calibration

When given a discrete target loss, such as for classification-like problems, direct empirical risk minimization is typically NP-hard, forcing one to find a more tractable surrogate. To ensure consistency, the literature has embraced the notion of *calibration* from Steinwart and Christmann [40, Chapter 3], which aligns with the definition in Tewari and Bartlett [41] for multiclass classification, and its generalizations to arbitrary discrete target losses [2, 33]. Calibration is more tractable and weaker than consistency, yet the two are equivalent under suitable assumptions [33, 41], notably in Quadrant 1. Intuitively, calibration says one cannot achieve the optimal surrogate loss while linking to a suboptimal target prediction.

**Definition 11** (Calibrated: Quadrant 1). *Let $\ell : \mathcal{R} \times \mathcal{Y} \to \mathbb{R}$ be a discrete target loss. A surrogate loss $L : \mathbb{R}^d \times \mathcal{Y} \to \mathbb{R}$ and link $\psi : \mathbb{R}^d \to \mathcal{R}$ pair $(L, \psi)$ is $\mathcal{P}$-calibrated with respect to $\ell$ if*

$$\forall p \in \mathcal{P} : \inf_{u \in \mathbb{R}^d : \psi(u) \notin \arg\min_r \mathbb{E}_p \ell(r, Y)} \mathbb{E}_p L(u, Y) > \inf_{u \in \mathbb{R}^d} \mathbb{E}_p L(u, Y) . \tag{7}$$

*We simply say $L$ is calibrated if $\mathcal{P} = \Delta_{\mathcal{Y}}$.*

Many works characterize calibrated surrogates for specific discrete target losses [3, 28, 41, 44], including the canonical 0-1 loss for binary and multiclass classification. We give another definition of calibration which is a special case of calibration via Steinwart and Christmann [40], and show it is equivalent to Definition 11 in discrete prediction settings, but can be applied in continuous estimation settings as well. We use this more general definition of calibration when proving statements about the relationship between consistency, calibration, and indirect elicitation.

The close connection between indirect elicitation and consistency was first explored by Agarwal and Agarwal [2]. In particular, calibration of $L \in \mathcal{L}$ with respect to $\ell$ implies indirect elicitation quite directly: take $u \in \mathbb{R}^d$ and $p \in \Gamma_u$, implying $u \in \Gamma(p)$. From eq. (1), $\mathbb{E}_p L(u, Y) = \inf_{u' \in \mathbb{R}^d} \mathbb{E}_p L(u', Y)$, so we must have $\psi(u) \in \gamma(p)$ from eq. (7), as desired.

For a given $p \in \mathcal{P}$, the (conditional) *regret*, or excess risk, of a loss $L$ is given by $R_L(u, p) := \mathbb{E}_p L(u, Y) - \inf_{u^*} \mathbb{E}_p L(u^*, Y)$.

**Definition 12** (Calibrated: Quadrants 1 and 3). *A loss $L : \mathbb{R}^d \times \mathcal{Y} \to \mathbb{R}$ is $\mathcal{P}$-calibrated with respect to a loss $\ell : \mathcal{R} \times \mathcal{Y} \to \mathbb{R}$ if there is a link $\psi : \mathbb{R}^d \to \mathcal{R}$ such that, for all distributions $p \in \mathcal{P}$, there exists a function $\zeta : \mathbb{R}_+ \to \mathbb{R}_+$ with $\zeta$ continuous at $0^+$ and $\zeta(0) = 0$ such that for all $u \in \mathbb{R}^d$, we have*

$$\ell(\psi(u); p) - \underline{\ell}(p) \le \zeta \left( \mathbb{E}_p L(u, Y) - \underline{L}(p) \right) . \tag{8}$$

*If $\mathcal{P} = \Delta_{\mathcal{Y}}$, we simply say $(L, \psi)$ is calibrated.*

Consider the following four conditions: Suppose we are given $\zeta : \mathbb{R}_+ \to \mathbb{R}_+$.

    A $\zeta$ satisfies $\zeta : 0 \mapsto 0$ and is continuous at 0.

    B $\epsilon_m \to 0 \implies \zeta(\epsilon_m) \to 0$.

    C Given $\zeta : \mathbb{R} \to \mathbb{R}_+$, for all $u \in \mathbb{R}^d$, $R_\ell(\psi(u); p) \le \zeta(R_L(u; p))$.

    D For all $p \in \mathcal{P}$ and sequences $\{u_m\}$ so that $R_L(u_m; p) \to 0$, we have $R_\ell(\psi(u_m); p) \to 0$.

The existence of a function $\zeta$ so that $(A \wedge C)$ defines calibration as in Definition 12, and we show $A \iff B$ in Lemma 4. Lemma 5 shows calibration if and only if $D$, which yields a condition equivalent to calibration without dependence the function $\zeta$.

**Proposition 3.** *When $\mathcal{R}$ and $\mathcal{Y}$ are finite, a continuous loss and link $(L, \psi)$ are $\mathcal{P}$-calibrated with respect to a target loss $\ell$ via Definition 12 if and only if they are $\mathcal{P}$-calibrated via Definition 11.*

*Proof.* $\implies$ We prove the contrapositive; if $(L, \psi)$ is not calibrated with respect to $\ell$ by Definition 11, then it is not calibrated via Definition 12 either. If $(L, \psi)$ are not calibrated with respect to $\ell$ by Definition 11, then there is a $p \in \mathcal{P}$ so that $\inf_{u : \psi(u) \notin \gamma(p)} \mathbb{E}_p L(u, Y) = \inf_u \mathbb{E}_p L(u, Y)$. Thus there is a sequence $\{u_m\}$ so that $\lim_{m \to \infty} \psi(u_m) \notin \gamma(p)$ and

$\mathbb{E}_p L(u_m, Y) \to \underline{L}(p)$. Now we have $R_L(u_m; p) \to 0$ but $R_\ell(\psi(u_m); p) \not\to 0$, so by Lemma 5, we contradict calibration by Definition 12.

$\impliedby$ Suppose there was a function $\zeta$ satisfying the bound in eq. (8) for a fixed distribution $p \in \mathcal{P}$. Observe the bound in eq. (7) can be written as $R_L(u, p) > 0$ for all $p \in \Delta_\mathcal{Y}$ and $u$ such that $\psi(u) \neq \gamma(p)$. By eq. (8), for any sequence $\{u_m\}$ so that $\psi(u_m) \not\to \gamma(p)$, we have must have $\zeta(R_\ell(\psi(u_m), p)) \not\to 0$ as we would otherwise contradict the bound in eq. (8) since $R_\ell(\psi(u), p) \not\to 0$. Therefore $R_L(u_m, p) \not\to 0$; thus, the strict inequality holds. $\qquad\square$

The following Lemma shows that conditions $A$ and $B$ are equivalent, so that we can using condition $B$ in lieu of condition $A$ in the proof of Lemma 5

**Lemma 4.** *A function $\zeta : \mathbb{R} \to \mathbb{R}$ is continuous at 0 and $\zeta(0) = 0$ if and only if the sequence $\{u_m\} \to 0 \implies \zeta(u_m) \to 0$.*

*Proof.* $\implies$ Suppose we have a sequence $\{u_m\} \to 0$. By continuity, we have $\lim_{u_m \to 0} \zeta(u_m) = \zeta(0) = 0$, so $\zeta(u_m) \to 0$.

$\impliedby$ Suppose $\zeta(0) \neq 0$ but $\zeta$ was continuous at 0. The constant sequence $\{u_m\} = 0$ then converges to 0, but as $\zeta$ is continuous at 0, we must have $\lim_{m \to \infty} \zeta(u_m) = \zeta(0) \neq 0$, so $\zeta(u_m) \not\to 0$.

Now suppose $\zeta(0) = 0$ but $\zeta$ was not continuous at 0. There must be a sequence $\{u_m\} \to 0$ so that $\lim_{m \to \infty} \zeta(u_m) \neq \zeta(0) = 0$, so $\zeta(u_m) \not\to 0$. $\qquad\square$

Lemma 5 now gives a condition equivalent to calibration without requiring one to already have a function $\zeta$ in mind.

**Lemma 5.** *A continuous surrogate and link $(L, \psi)$ are $\mathcal{P}$-calibrated (via definition 12) with respect to $\ell$ if and only if, for all $p \in \mathcal{P}$ and sequences $\{u_m\}$ so that $R_L(u_m; p) \to 0$, we have $R_\ell(\psi(u_m); p) \to 0$.*

*Proof.* $\implies$ Take a sequence $\{u_m\}$ so that $R_L(u_m; p) \to 0$. Since $\zeta(0) = 0$ and $\zeta$ is continuous at 0, we have $\zeta(R_L(u_m; p)) \to 0$. As the bound from Equation (8) is satisfied for all $u \in \mathbb{R}^d$ by assumption, we observe

$$\forall m, 0 \leq R_\ell(\psi(u_m); p) \leq \zeta(R_L(u_m; p))$$
$$\implies 0 \leq \lim_{m \to \infty} R_\ell(\psi(u_m); p) \leq \lim_{m \to \infty} \zeta(R_L(u_m; p)) = 0$$
$$\implies 0 = \lim_{m \to \infty} R_\ell(\psi(u_m); p) .$$

$\impliedby$ Fix $p \in \mathcal{P}$, and consider $\zeta(c) := \sup_{u : R_L(u, p) \leq c} R_\ell(\psi(u); p)$. We will show $R_L(u_m; p) \to 0 \implies R_\ell(\psi(u_m); p) \to 0$ gives calibration via the function $\zeta$ constructed above. With $\zeta$ as constructed, we observe that the bound in equation (8) is satisfied for all $u \in \mathbb{R}^d$ and apply Lemma 4 to observe that if there is a sequence $\{\epsilon_m\} \to 0$ so that $\zeta(\epsilon_m) \not\to 0$, it is because $R_L(u_m, p) \not\to 0 \implies R_\ell(\psi(u_m), p) \to 0$.

Now, we observe that the bound in Equation (8) is satisfied for all $u \in \mathbb{R}^d$ by construction of $\zeta$. Let $S(v) := \{u' \in \mathbb{R}^d : R_L(u'; p) \leq R_L(v, p)\}$. Showing $R_\ell(\psi(u); p) \leq \sup_{u' \in S(u)} R_\ell(\psi(u'); p)$ for all $u \in \mathbb{R}^d$ gives the condition $C$. As $u$ is in the space over which the supremum is being taken (as $R_L(u; p) \leq R_L(u; p)$), we then have calibration by definition of the supremum.

Now suppose there exists a sequence $\{\epsilon_m\} \to 0$ so that $\zeta(\epsilon_m) \not\to 0$. Consider $S(\epsilon) = \{u \in \mathbb{R}^d : R_L(u, p) \leq \epsilon\}$.

$$\epsilon_1 \leq \epsilon_2 \implies S(\epsilon_1) \subseteq S(\epsilon_2)$$
$$\implies \zeta(\epsilon_1) \leq \zeta(\epsilon_2) .$$

Now suppose there exists a sequence $\{u_m\}$ so that $R_L(u_m, p) \to 0$. Then for all $\epsilon > 0$, there exists a $m' \in \mathbb{N}$ so that $R_L(u_m, p) < \epsilon$ for all $m \geq m'$. Since this is true for all $\epsilon$, we have $S(\epsilon)$ nonempty for all $\epsilon > 0$, and therefore $\zeta(c)$ is discrete for all $c > 0$. Now if $\zeta(\epsilon_m) \not\to 0$, it

must be because $R_\ell(\psi(u_m), p) \not\to 0$ for some sequence converging to zero surrogate regret, and therefore we contradict the statement $R_L(u_m, p) \to 0 \implies R_\ell(\psi(u_m), p) \to 0$.

Moreover, we argue that such a sequence of $\{u_m\}$ with converging surrogate regret always exists by continuity and boundedness from below of the surrogate loss, since we can take the constant sequence at the (attained) infimum. $\qquad\square$

### C.1 Relating calibration, consistency, and indirect elicitation.

Even with the more general notion of calibration that extends beyond discrete predictions, we still have consistency implying calibration.

**Proposition 4.** *If a loss and link $(L, \psi)$ are consistent with respect to a loss $\ell$, then they are calibrated with respect to $\ell$.*

*Proof.* We show the contrapositive. If $(L, \psi)$ are not calibrated with respect to $\ell$, then there is a sequence $\{u_m\}$ such that $R_L(u_m; p) \to 0$ but $R_\ell(\psi(u_m); p) \not\to 0$ via Lemma 5. Suppose $D \sim \mathcal{X} \times \mathcal{Y}$ has only one $x \in \mathcal{X}$ with $Pr_D(X = x) > 0$ so that $p := D_x$ and $\mathbb{E}_D f(X, Y) = \mathbb{E}_p f(x, Y)$. Consider any sequence of functions $\{f_m\} \to f$ with $f_m(x) = u_m$ for all $f_m$. Now we have $\mathbb{E}_D L(f_m(X), Y) \to \inf_f \mathbb{E}_D L(f(X), Y)$, but $\mathbb{E}_D \ell(\psi \circ f(X), Y) \not\to \inf_f \mathbb{E}_D \ell(\psi \circ f(X), Y)$, and therefore $(L, \psi)$ is not consistent with respect to $\ell$. $\qquad\square$

Moreover, we have calibration implying indirect elicitation.

**Lemma 6.** *If a surrogate and link $(L, \psi)$ with $L \in \mathcal{L}$ are calibrated with respect to a loss $\ell : \mathcal{R} \times \mathcal{Y} \to \mathbb{R}$, then $L$ indirectly elicits the property $\gamma := \mathrm{prop}_{\mathcal{P}}[\ell]$.*

*Proof.* Let $\Gamma$ be the unique property directly elicited by $L$, and fix $p \in \Delta_\mathcal{Y}$ with $u$ such that $p \in \Gamma_u$. We know such a $u$ exists since $\Gamma(p) \neq \emptyset$. As $p \in \Gamma_u$, then $\zeta(\mathbb{E}_p L(u, Y) - \underline{L}(p)) = \zeta(0) = 0$, we observe the bound $\ell(\psi(u); p) \leq \underline{\ell}(p)$. We also have $\ell(\psi(u); p) \geq \underline{\ell}(p)$ by definition of $\underline{\ell}$, so we must have $\ell(\psi(u); p) = \underline{\ell}(p) = \ell(\gamma(p); p)$, and therefore, $p \in \gamma_{\psi(u)}$. Thus, we have $\Gamma_u \subseteq \gamma_{\psi(u)}$, so $L$ indirectly elicits $\gamma$. $\qquad\square$

Combining the two results, we can observe the result of Proposition 1 another way: *through calibration.*

## D  Quadrant 1: Previous Lower Bounds and Comparisons

The main known technique for lower bounds on surrogate dimensions is given by Ramaswamy and Agarwal [33] for the Quadrant 1 (target loss and discrete predictions). The proof heavily builds around the "limits of sequences" in the definition of calibration. By restricting slightly to the broad class of minimizable losses $\mathcal{L}^{\mathrm{cvx}}$, we show their bound follows relatively directly from Corollary 3. (We conjecture that the minimizability restriction to $\mathcal{L}^{\mathrm{cvx}}$ can be lifted; see § 5.) Ramaswamy and Agarwal [33] construct what they call the subspace of feasible dimensions and give bounds in terms of its dimension.

**Definition 13** (Subspace of feasible directions). *The subspace of feasible directions $\mathcal{S}_\mathcal{C}(p)$ of a convex set $\mathcal{C} \subseteq \mathbb{R}^n$ at $p \in \mathcal{C}$ is $\mathcal{S}_\mathcal{C}(p) = \{v \in \mathbb{R}^n : \exists \epsilon_0 > 0 \text{ such that } p + \epsilon v \in \mathcal{C} \ \forall \epsilon \in (-\epsilon_0, \epsilon_0)\}$.*

Ramaswamy and Agarwal [33] gives a lower bound on the dimensionality of all consistent convex surrogates, i.e. $\mathrm{cons}_{\mathrm{cvx}}(\ell) \geq \|p\|_0 - \dim(\mathcal{S}_{\gamma_r}(p)) - 1$ for all $p$ and $r \in \gamma(p)$, particularly in the setting where one is given a discrete prediction problem and target loss over finite outcomes. It turns out that the subspace of feasible directions is essentially a special case of a flat described by Theorem 1. So, by making a slight restriction to the class of minimizable convex surrogates $\mathcal{L}^{\mathrm{cvx}}$, we can derive this lower bound from our general technique in a way that we find shorter and simpler.

**Corollary 6** ([33] Theorem 18). *Let $\ell : \mathcal{R} \times \mathcal{Y} \to \mathbb{R}$ be a discrete loss eliciting $\gamma : \Delta_\mathcal{Y} \rightrightarrows \mathcal{R}$ with $\mathcal{Y}$ finite. Then for all $p \in \Delta_\mathcal{Y}$ and $r \in \gamma(p)$,*

$$\mathrm{cons}_{\mathrm{cvx}}(\gamma) \geq \|p\|_0 - \dim(\mathcal{S}_{\gamma_r}(p)) - 1 \ . \tag{9}$$

*Sketch.* If $\text{cons}_{\text{cvx}}(\gamma) \leq d$, then there is a $L \in \mathcal{L}_d^{\text{cvx}}$ so that $L$ is consistent with respect to $\gamma$, and in turn, indirectly elicits $\gamma$. Theorem 1 says that there is some $d$-flat $F = \ker_{\mathcal{P}} V$ such that $p \in F \subseteq \gamma_r$. In particular, if $p \in \text{relint}(\Delta_{\mathcal{Y}})$, we can see $\dim(F) = \dim(\mathcal{S}_{\gamma_r}(p))$. Since affhull($\Delta_{\mathcal{Y}}$) has dimension $|\mathcal{Y}| - 1 = \|p\|_0 - 1$, by rank-nullity and $\text{rank}(V) \leq d$ (more precisely, the corresponding linear map $q \mapsto \mathbb{E}_q V$) we have $d \geq \|p\|_0 - 1 - \dim(\mathcal{S}_{\gamma_r}(p))$.

When $p \notin \text{relint}(\Delta_{\mathcal{Y}})$, we can project down to the subsimplex on the support of $p$, again of dimension $\|p\|_0 - 1$, and modify $L$ and $\ell$ accordingly. Now $p$ is in the relative interior of this subsimplex, so the above gives $\text{cons}_{\text{cvx}}(\gamma) \geq \|p\|_0 - 1 - \dim(\mathcal{S}_{\gamma_r}(p))$, where now $\mathcal{S}$ is relative to $\mathbb{R}^{\text{supp}(p)}$. Finally, the feasible subspace dimension in the projected space is the same as in the original space because of $p$'s location on a face of $\Delta_{\mathcal{Y}}$. $\qquad\square$

There are some cases where the bound provided by Corollaries 2 and 3 is strictly tighter than the bound provided by feasible subspace dimension in Corollary 6. For an example of how Corollary 2 applies to a discrete property for which there is no target loss − a non-elicitable property, i.e. Quadrant 2, which is not considered by Ramaswamy et al. [34] − we refer the reader to Figure 3.

**Example: Abstain** Recall the abstain target loss $\ell^{abs}(r, y) := \mathbf{I}\{r \notin \{y, \bot\}\} + (1/2)\mathbf{I}\{r = \bot\}$, we can consider the *abstain property* it elicits, where one predicts the most likely outcome $y$ if $Pr[Y = y|x] \geq 1/2$ and "abstain" by predicting $\bot$ otherwise. Ramaswamy and Agarwal [33] present a convex surrogate for the abstain loss that takes as input a prediction whose dimension is logarithmic in the number of outcomes, yielding new upper bounds on $\text{cons}_{\text{cvx}}(\ell^{abs})$ which are an exponential improvement over previous results, e.g., [9].

To lower bound the dimension of convex surrogates, we can consider two different distributions; in the first, our bound yields a strict gap over the feasible subspace dimension bound, and in the second, the bounds are equal. First, we choose $p = \bullet$ to be the uniform distribution (see Figure 4). In this case, the bound by feasible subspace dimension yields $\text{cons}_{\text{cvx}}(\ell^{abs}) \geq 3 - 2 - 1 = 0$, as the feasible subspace dimension is 2 since we are on the relative interior of the level set and simplex, as shown in Figure 4 (L).

However, consider any 1-flat containing $\bullet$. When intersected with the simplex, one can see that any line (a 1-flat, since $\bullet \in \text{relint}(\Delta_{\mathcal{Y}})$) in the simplex through $\bullet$ also leaves the cell $\gamma_{\bot}$, which contains $p$. See Figure 4 (R) for intuition; a 1-flat through $p \in \text{relint}(\Delta_{\mathcal{Y}})$ would be a line in such a figure. Therefore, we have no 1-flat containing $p$ staying in $\gamma_{\bot}$, so we obtain a better lower bound, $\text{cons}_{\text{cvx}}(\ell^{abs}) \geq 2$. Combining this with the upper bounds given by [34], we observe the bound $\text{cons}_{\text{cvx}}(\ell^{abs}) = 2$ is tight in this case with $|\mathcal{Y}| = 3$.

Our bounds sometimes match those of [33]; consider the distribution $\star = (1/4, 1/4, 1/2)$, shown in Figure 4. The feasible subspace dimension of both $\gamma_{\bot}$ and $\gamma_3$ at $\star$ is 1, since one only moves toward the distributions $(0, 1/2, 1/2)$ and $(1/2, 0, 1/2)$ without leaving the level sets, and the three points are collinear in affhull($\Delta_{\mathcal{Y}}$), suggesting $\mathcal{S}_{\gamma_{\bot}}(q) = 1$. This yields $\text{cons}_{\text{cvx}}(\ell^{abs}) \geq 3 - 1 - 1 = 1$. The same line segment defines a flat contained in both $\gamma_{\bot}$ and $\gamma_3$, so we have $\text{cons}_{\text{cvx}}(\ell^{abs}) \geq 1$ by Corollary 3, matching the feasible subspace dimension bound.

Bounds using $d$-flats appear to work well at distributions where previous bounds via feasible subspace dimension would have been vacuous. In essence, flats allow us a "global" view of the property we are eliciting, while the feasible subspace method only permits a "local" look at the property, so we find our method works better for distributions in $\text{relint}(\Delta_{\mathcal{Y}})$.

### D.1  Reconstructing Ramaswamy and Agarwal [33, Thm. 16]

**Lemma 7.** *Let the $d$-flat $F \subseteq \mathcal{P}$ (defined over finite $\mathcal{Y}$) contain some $p \in \text{relint}(\mathcal{P})$. Then*

*(i) $p \in \text{relint}(F)$;*

*(ii) $\dim(\mathcal{S}_F(p)) \geq \dim(\text{affhull}(\mathcal{P})) - d$.*

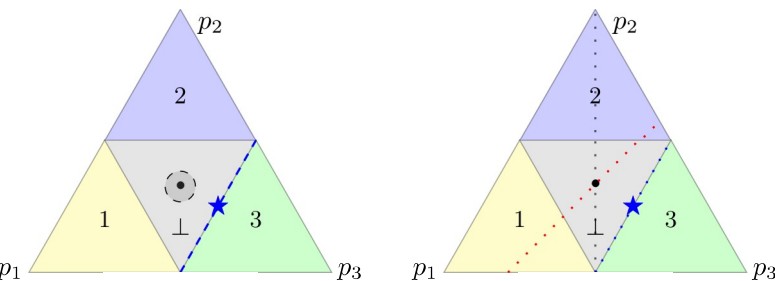

Figure 4: (Left) Feasible subspace dimension $\mathcal{S}_{\gamma_\perp}(\bullet) = 2$ and $\mathcal{S}_{\gamma_\perp}(\star) = 1$, giving the bound $\mathrm{cons}_{\mathrm{cvx}}(\ell^{abs}) \geq 3 - 1 - 1 = 1$. (Right) No 1-flat through $\bullet$ (a line since $\bullet \in \mathrm{relint}(\Delta_{\mathcal{Y}})$) stays fully contained in $\gamma_\perp$, so $\mathrm{cons}_{\mathrm{cvx}}(\ell^{abs}) \geq 2$.

*Proof.* As $F$ is a $d$-flat, we have some $W : \mathcal{Y} \to \mathbb{R}^d$ such that $F = \ker_{\mathcal{P}} W$. Throughout, given a point (typically a distribution) $p$ and convex set $P$, we define $P_p := P - \{p\}$. Define $T_W : \mathrm{span}(\mathcal{P}_p) \to \mathbb{R}^d, v \mapsto \mathbb{E}_v W$.

(i) Since $p \in \mathrm{relint}(\mathcal{P})$, for all $q \in \mathcal{P}$, there is some small enough $\epsilon > 0$ so that for all $\alpha \in (-\epsilon, \epsilon)$, the point $q_\alpha := p - \alpha(q - p)$ is still in $\mathcal{P}$. In particular, for $q \in F$, we claim $q_\alpha \in F$. As $p, q \in F$, we have $\mathbb{E}_p W = \mathbb{E}_q W = \vec{0}$. By linearity of expectation, we then have $\mathbb{E}_{q_\alpha} W = \vec{0}$. This implies $q_\alpha \in F$, and therefore $p \in \mathrm{relint}(F)$.

(ii) We first show $\mathrm{span}(F_p) = \mathcal{S}_F(p)$. First, take $v \in \mathcal{S}_F(p)$, and take $\epsilon_0$ as in the definition. For $\epsilon = \epsilon_0/2$, we then have $p + \epsilon v \in F \implies \epsilon v \in F_p$, and therefore, $v \in \mathrm{span}(F_p)$. Now take $v \in \mathrm{span}(F_p)$. Since $p \in \mathrm{relint}(F)$ (i), we have $\vec{0} \in \mathrm{relint}(F_p)$. Therefore there is an $\epsilon_0 > 0$ so that $\epsilon v \in F_p$ for all $\epsilon \in (-\epsilon_0, \epsilon_0)$ by convexity of $F$. Therefore, $v \in \mathcal{S}_F(p)$, and we observe $\mathcal{S}_F(p) = \mathrm{span}(F_p)$.

We now show $\mathcal{S}_F(p) = \ker(T_W)$. Observe that $\mathcal{S}_F(p) \subseteq \ker(T_W)$ follows trivially from the definitions of the two functions. Now let $v \in \ker(T_W)$, and $v' \in F_p$. This means $\mathbb{E}_v W = \vec{0}$, so it suffices to show $v = cv' \in F_p$, thus showing $v \in \mathcal{S}_F(p)$. Since $p \in \mathrm{relint}(\mathcal{P})$, we must have $\vec{0} \in \mathrm{relint}(F_p)$, so we know there is some small enough $\epsilon > 0$ so that $-\alpha v' \in F_p$ for $\alpha \in (-\epsilon, \epsilon)$. Take $c = -\alpha$, and we conclude $v \in \mathcal{S}_F(p)$. Therefore, $\ker(T_W) = \mathcal{S}_F(p)$.

We finally want to show $\dim(\mathrm{affhull}(\mathcal{P})) = \dim(\mathrm{span}(\mathcal{P}_p))$. Consider that any $q \in \mathrm{span}(\mathcal{P}_p)$ can be written as a scalar multiple of an element of $\mathcal{P}_p$, which can be written as a convex combination of elements of the minimal basis $\mathcal{P}_p$. In particular, since $\vec{0} \in \mathcal{P}_p$, it can be written as an affine combination of elements of the basis, so $\dim(\mathrm{affhull}(\mathcal{P})) \geq \dim(\mathrm{span}(\mathcal{P}_p))$. We also have $\mathrm{affhull}(\mathcal{P}) - \{p\} \subseteq \mathrm{span}(\mathcal{P}_p)$, so $\dim(\mathrm{affhull}(\mathcal{P})) = \dim(\mathrm{affhull}(\mathcal{P}) - \{p\}) \leq \mathrm{span}(\mathcal{P}_p)$. Therefore, $\dim(\mathrm{affhull}(\mathcal{P})) = \dim(\mathrm{span}(\mathcal{P}_p))$.

As $\mathcal{Y}$ is a finite set, $\mathrm{span}(\mathcal{P}_p)$ is a finite-dimensional vector space. The rank-nullity theorem states $\dim(\mathrm{im}(T_W)) + \dim(\ker(T_W)) = \dim(\mathrm{span}(\mathcal{P}_p)) = \dim(\mathrm{affhull}(\mathcal{P}))$. As $\dim(\mathrm{im}(T_W)) \leq d$, and we have shown above that $\mathcal{S}_F(p) = \mathrm{span}(F_p) = \ker(T_W)$, the conclusion follows. $\square$

**Corollary 6** ([33] Theorem 18). *Let $\ell : \mathcal{R} \times \mathcal{Y} \to \mathbb{R}$ be a discrete loss eliciting $\gamma : \Delta_{\mathcal{Y}} \rightrightarrows \mathcal{R}$ with $\mathcal{Y}$ finite. Then for all $p \in \Delta_{\mathcal{Y}}$ and $r \in \gamma(p)$,*

$$\mathrm{cons}_{\mathrm{cvx}}(\gamma) \geq \|p\|_0 - \dim(\mathcal{S}_{\gamma_r}(p)) - 1 \ . \tag{9}$$

*Proof.* Let $L \in \mathcal{L}_d^{\mathrm{cvx}}$ be a calibrated surrogate for $\ell$, and let $\Gamma := \mathrm{prop}_{\Delta_{\mathcal{Y}}}[L]$. Consider $\mathcal{Y}' := \{y \in \mathcal{Y} : p_y > 0\}$ and $p' = (p_y)_{y \in \mathcal{Y}'} \in \Delta_{\mathcal{Y}'}$. Take $L' := L|_{\mathcal{Y}'}$ and $\ell' := \ell|_{\mathcal{Y}'}$. Define $h : \mathbb{R}^{\mathcal{Y}'} \to \mathbb{R}_{\mathcal{Y}}$ such that $h(q') = q$ such that $q_y = q'_y$ for $y \in \mathcal{Y}'$ and $q_y = 0$ otherwise. Take $\Gamma' = \Gamma \circ h$, $\gamma' = \gamma \circ h$.

We wish to first show $L'$ indirectly elicits $\gamma'$. Since $L$ indirectly elicits $\gamma$, we have a link $\psi$ such that for all $u \in \mathbb{R}^d$, $\Gamma_u \subseteq \gamma_{\psi(u)}$. As $\Gamma'(q) = \Gamma(h(q))$ and $\gamma'(q) = \gamma(h(q))$, we have

$q \in \Gamma'_u \iff h(q) \in \Gamma_u \implies h(q) \in \gamma_{\psi(u)} \iff (q_y)_{y \in \mathcal{Y}'} \in \gamma'_{\psi(u)}$, and therefore, $L'$ indirectly elicits $\gamma'$ via the link $\psi \circ \text{proj}(\mathcal{Y}')$, where $\text{proj}(\mathcal{Y}') : q \mapsto (q_y)_{y \in \mathcal{Y}'}$.

We aim to show $\dim(\mathcal{S}_{\gamma_r}(p)) \geq \dim(\mathcal{S}_{\gamma'_r}(p'))$. We do this by showing that $h(\mathcal{S}_{\gamma'_r}(p')) \subseteq \mathcal{S}_{\gamma_r}(p)$, and the result holds as $h$ is linear and injective. Suppose $v \in h(\mathcal{S}_{\gamma'_r}(p'))$, then there exists a $v'$ so that $v = h(v')$ and an $\epsilon_0 > 0$ such that $\epsilon v' + p' \in \gamma'_r$ for all $\epsilon \in (-\epsilon_0, \epsilon_0)$. Since $h$ is linear and recall $h(\gamma'_r) \subseteq \gamma_r$, this implies $\epsilon v + p \in \gamma_r$ for all $\epsilon \in (-\epsilon_0, \epsilon_0)$. Therefore $v \in \mathcal{S}_{\gamma_r}(p)$, and the result follows.

As $L'$ indirectly elicits $\gamma'$, by Corollary 3, we know there exists a $d$-flat $F$ with $p' \in F \subseteq \gamma'_r$. Taking $\mathcal{P} = \Delta_{\mathcal{Y}'}$, we know $p' \in \text{relint}(\Delta_{\mathcal{Y}'})$ by construction, so we can apply Lemma 7(ii), which gives $\dim(\mathcal{S}_F(p')) \geq \dim(\text{affhull}(\Delta_{\mathcal{Y}'})) - d = \|p\|_0 - 1 - d$.[6] Additionally, $\mathcal{S}_F(p') \subseteq \mathcal{S}_{\gamma'_r}(p')$ by subset inclusion of the sets themselves. Chaining these results, we obtain

$$\dim(\mathcal{S}_{\gamma_r}(p)) \geq \dim(\mathcal{S}_{\gamma'_r}(p')) \geq \dim(\mathcal{S}_F(p')) \geq \|p\|_0 - 1 - d .$$

$\square$

# E   Proof of Theorem 2

Throughout this section, we will assume $\mathcal{P}$ is convex. See Frongillo and Kash [20, §E.5] for a discussion of how to relax this assumption.

## E.1   General setting of elicitation complexity

We briefly introduce the general notion of elicitation complexity, of which Definition 6 is a special case, as some statements are more naturally made in this general setting.

**Definition 14.** $\Gamma'$ refines $\Gamma$ *if for all* $r' \in \text{range}\,\Gamma'$ *there exists* $r \in \text{range}\,\Gamma$ *with* $\Gamma'_{r'} \subseteq \Gamma_r$.

Equivalently, $\Gamma'$ refines $\Gamma$ if there is a link function $\psi : \text{range}\,\Gamma' \to \text{range}\,\Gamma$ such that $\Gamma'_{r'} \subseteq \Gamma_{\psi(r')}$ for all $r' \in \text{range}\,\Gamma'$.

**Definition 15.** *For* $k \in \mathbb{N} \cup \{\infty\}$, *let* $\mathcal{E}_k(\mathcal{P})$ *denote the class of all elicitable properties* $\Gamma : \mathcal{P} \to \mathbb{R}^k$, *and* $\mathcal{E}(\mathcal{P}) := \bigcup_{k \in \mathbb{N} \cup \{\infty\}} \mathcal{E}_k(\mathcal{P})$. *When* $\mathcal{P}$ *is implicit we simply write* $\mathcal{E}$.

**Definition 16.** *Let* $\mathcal{C}$ *be a class of properties. The* elicitation complexity *of a property* $\Gamma$ *with respect to* $\mathcal{C}$, *denoted* $\text{elic}_{\mathcal{C}}(\Gamma)$, *is the minimum value of* $k \in \mathbb{N} \cup \{\infty\}$ *such that there exists* $\hat{\Gamma} \in \mathcal{C} \cap \mathcal{E}_k(\mathcal{P})$ *that refines* $\Gamma$.

## E.2   Supporting statements

**Proposition 5** (Osband [31]). *Let* $\Gamma$ *be elicitable. Then* $\Gamma_r$ *is convex for all* $r \in \text{range}\,\Gamma$.

**Lemma 8** (Set-valued extension of Frongillo and Kash [20, Lemma 4]). *If* $\Gamma'$ *refines* $\Gamma$ *then* $\text{elic}_{\mathcal{C}}(\Gamma') \geq \text{elic}_{\mathcal{C}}(\Gamma)$.

*Proof.* As $\Gamma'$ refines $\Gamma$, we have some $\psi : \text{range}\,\Gamma' \to \text{range}\,\Gamma$ such that for all $r' \in \text{range}\,\Gamma'$ we have $\Gamma'_{r'} \subseteq \Gamma_{\psi(r')}$. Suppose we have $\hat{\Gamma} \in \mathcal{C}$ and $\varphi : \text{range}\,\hat{\Gamma} \to \text{range}\,\Gamma'$ such that for all $u \in \text{range}\,\hat{\Gamma}$ we have $\hat{\Gamma}_u \subseteq \Gamma'_{\varphi(u)}$. Then for all $u \in \text{range}\,\hat{\Gamma}$ we have $\hat{\Gamma}_u \subseteq \Gamma'_{\varphi(u)} \subseteq \Gamma_{(\psi \circ \varphi)(u)}$. In particular, if $\text{elic}_{\mathcal{C}}(\Gamma') = m$, then we have such a $\hat{\Gamma} : \mathcal{P} \rightrightarrows \mathbb{R}^m$, and hence $\text{elic}_{\mathcal{C}}(\Gamma) \leq m$. $\square$

**Lemma 9** (Frongillo and Kash [20, Lemma 8]). *Suppose* $L \in \mathcal{L}$ *elicits* $\Gamma : \mathcal{P} \to \mathcal{R}$ *and has Bayes risk* $\underline{L}$. *Then for any* $p, p' \in \mathcal{P}$ *with* $\Gamma(p) \neq \Gamma(p')$, *we have* $\underline{L}(\lambda p + (1 - \lambda)p') > \lambda \underline{L}(p) + (1 - \lambda)\underline{L}(p')$ *for all* $\lambda \in (0, 1)$.

**Lemma 10** (Adapted from Frongillo and Kash [20, Theorem 4]). *If* $L$ *elicits a single-valued* $\Gamma$, *and* $\hat{\Gamma}$ *refines* $\underline{L}$, *then* $\hat{\Gamma}$ *refines* $\Gamma$.

---

[6]To reason about $\dim(\text{affhull}(\Delta_{\mathcal{Y}'})) = \|p\|_0 - 1$, observe that the uniform distribution on $\Delta_{\mathcal{Y}'}$ has full support and therefore requires $\|p\|_0 - 1$ elements in its basis.

*Proof.* Suppose for a contradiction that $\hat\Gamma$ does not refine $\Gamma$. Then we have some $u \in \text{range}\,\hat\Gamma$ such that for all $r \in \text{range}\,\Gamma$ we have $\hat\Gamma_u \not\subseteq \Gamma_r$. In particular, recalling that $\Gamma$ is single-valued, we must have $p, p' \in \hat\Gamma_u$ such that $\Gamma(p) \neq \Gamma(p')$. Moreover, as $\hat\Gamma$ refines $\underline{L}$, we also have $\underline{L}(p) = \underline{L}(p')$. From Lemma 9 and $\lambda = 1/2$ we have $\underline{L}(q) > \frac{1}{2}\underline{L}(p) + \frac{1}{2}\underline{L}(p') = \underline{L}(p)$, where $q = \frac{1}{2}p + \frac{1}{2}p'$. As the level set $\hat\Gamma_u$ is convex by Proposition 5, we also have $q \in \hat\Gamma_u$, and hence $\underline{L}(q) = \underline{L}(p)$, a contradiction. $\qquad\square$

**Lemma 11** (Minor modifications from Frongillo and Kash [20]). *Let $\mathcal{V}$ be a real vector space. Let $f : \mathcal{V} \to \mathbb{R}^k$ be linear and $C \subseteq \mathcal{V}$ convex with $\text{span}\,C = \mathcal{V}$, and let $m \in \mathbb{N}$. Suppose that $0 \in \text{int}\,f(C)$, and for all $v \in S := C \cap \ker f$, there exists a linear $\hat f_v : \mathcal{V} \to \mathbb{R}^m$ with $v \in C \cap \ker \hat f_v \subseteq S$. Then $m \geq k$. If $m = k$, we additionally have $0 \in \text{int}\,\hat f_v(C)$ for some $v \in S$.*

*Proof.* The condition $0 \in \text{int}\,f(C)$ is equivalent to the existence of some $v_1, \dots v_{k+1} \in C$ such that $0 \in \text{int}\,\text{conv}\{f(v_i) : i \in \{1, \dots, k+1\}\}$. Let $\alpha_1, \dots, \alpha_{k+1} > 0$, $\sum_{i=1}^{k+1} \alpha_i = 1$, such that $\sum_{i=1}^{k+1} \alpha_i f(v_i) = 0$. As these are barycentric coordinates, this choice of $\alpha_i$ is unique, a fact which will be important later. We will take $v = \sum_{i=1}^{k+1} \alpha_i v_i$, an element of $C$ by convexity, and thus an element of $S$ as $f(v) = 0$.

Let $\hat f_v : \mathcal{V} \to \mathbb{R}^m$ be linear with $v \in \hat S := C \cap \ker \hat f_v \subseteq S$. Let $\beta_1, \dots, \beta_{k+1} \in \mathbb{R}$, $\sum_{i=1}^{k+1} \beta_i = 0$, such that $\sum_{i=1}^{k+1} \beta_i \hat f_v(v_i) = 0$. We will show that the $\beta_i$ must be identically zero, i.e. that $\{\hat f_v(v_i) : i \in \{1, \dots, k+1\}\}$ are affinely independent. By construction, $v' := \sum_{i=1}^{k+1} \beta_i v_i \in \ker \hat f_v$, and as $v \in \ker \hat f_v$, for all $\lambda > 0$ we have $v_\lambda := v + \lambda v' = \sum_{i=1}^{k+1} (\alpha_i + \lambda\beta_i) v_i \in \ker \hat f_v$. Taking $\lambda$ sufficiently small, we have $\gamma_i := \alpha_i + \lambda\beta_i > 0$ for all $i$, and $\sum_{i=1}^{k+1} \gamma_i = \sum_{i=1}^{k+1} \alpha_i + \lambda\sum_{i=1}^{k+1} \beta_i = 1$. By convexity of $C$, we have $v_\lambda \in C$. Now $v_\lambda \in C \cap \ker \hat f_v \subseteq S = C \cap \ker f$, and in particular $v_\lambda \in \ker f$. Thus, $f(v_\lambda) = \sum_{i=1}^{k+1} \gamma_i f(v_i) = 0$. By the uniqueness of barycentric coordinates, for all $i \in \{1, \dots, k+1\}$, we must have $\gamma_i = \alpha_i$ and thus $\beta_i = 0$, as desired.

As $\hat f_v(C)$ contains $k+1$ affinely independent points, we have $m \geq \dim \text{im}\hat f_v \geq k$. When $m = k$, by affine independence, the set $\text{conv}\{\hat f_v(v_i) : i \in \{1, \dots, k+1\}\}$ has dimension $k$ in $\mathbb{R}^k$. As $0 = \hat f_v(v) = \sum_{i=1}^{k+1} \alpha_i \hat f_v(v_i)$, and $\alpha_i > 0$ for all $i$, we conclude $0 \in \text{int}\,\text{conv}\{\hat f_v(v_i) : i \in \{1, \dots, k+1\}\} \subseteq \text{int}\,\hat f_v(C)$. $\qquad\square$

**Lemma 12** (Frongillo and Kash [20, Lemma 14]). *Let $\mathcal{V}$ be a real vector space. Let $f : \mathcal{V} \to \mathbb{R}^k$ be linear, $C \subseteq \mathcal{V}$ convex with $\text{span}\,C = \mathcal{V}$, and let $S = C \cap \ker f$. If $0 \in \text{int}\,f(C)$ then $\text{span}\,S = \ker f$.*

### E.3 Proving the lower bound for Bayes risks

Let $\mathcal{C}_d^*$ be the class of properties $\Gamma$ which are elicited by a convex loss $L \in \mathcal{L}_d^{\text{cvx}}$ for some $d \in \mathbb{N}$, and let $\mathcal{C}^* := \bigcup_{d \in \mathbb{N}} \mathcal{C}_d^*$. Then for all properties $\gamma$, if $\text{elic}_{\mathcal{C}^*}(\gamma) < \infty$, we have $\text{elic}_{\mathcal{C}^*}(\gamma) = \text{elic}_{\text{cvx}}(\gamma)$, a fact we use tacitly in the proof.

**Theorem 2.** *Let $\mathcal{P}$ be a convex set of Lebesgue densities supported on the same set for all $p \in \mathcal{P}$. Let $\Gamma : \mathcal{P} \to \mathbb{R}^d$ satisfy Condition 1 for some $r \in \mathbb{R}^d$. Let $L \in \mathcal{L}^{\text{cvx}}$ elicit $\Gamma$ such that $\underline{L}$ is non-constant on $\Gamma_r$. Then $\text{cons}_{\text{cvx}}(\underline{L}) \geq \text{elic}_{\text{cvx}}(\underline{L}) \geq d + 1$.*

*Proof.* Let $V : \mathcal{Y} \to \mathbb{R}^d$ and $r$ be given by the statement of the theorem and from Condition 1. Let $m = \text{elic}_{\mathcal{C}^*}(\underline{L})$, so that we have $\hat\Gamma \in \mathcal{C}_m^*$ which refines $\underline{L}$. By Lemma 10 we have $\hat\Gamma$ refines $\Gamma$.

We now establish the conditions of Lemma 11 for $C = \mathcal{P}$. Let $f : \text{span}\,\mathcal{P} \to \mathbb{R}^d$, $p \mapsto \mathbb{E}_p V$. From Condition 1, we have $0 \in \text{int}\,f(\mathcal{P})$ and $\ker f \cap \mathcal{P} = \ker_{\mathcal{P}} V = \Gamma_r$. Now let $p \in \Gamma_r$ be arbitrary, and take any $u \in \hat\Gamma(p)$. As $\Gamma$ is single-valued, $r \in \text{range}\,\Gamma$ is the unique value with $p \in \Gamma_r$. As $\hat\Gamma$ refines $\Gamma$, there exists $r' \in \text{range}\,\Gamma$ with $\hat\Gamma_u \subseteq \Gamma_{r'}$, and since $p \in \hat\Gamma_u$, we conclude $r' = r$ from the above. From Theorem 1, we have some $\hat V_{u,p}$ with

$p \in \ker_{\mathcal{P}} \hat{V}_{u,p} \subseteq \hat{\Gamma}_u \subseteq \Gamma_r = \ker_{\mathcal{P}} V$. Letting $\hat{f}_p : \operatorname{span}\mathcal{P} \to \mathbb{R}^d$, $p \mapsto \mathbb{E}_p \hat{V}_{u,p}$, we have now satisfied the conditions of Lemma 11. We conclude $m \geq d$, and moreover, if $m = d$, then there exists some $q \in \Gamma_r$ such that $0 \in \operatorname{int} \hat{f}_q(\mathcal{P})$.

Now suppose $m = d$ for a contradiction. Let $\hat{S} := \ker f_q \cap \mathcal{P}$. Applying Lemma 12 to the functions $f$ and $\hat{f}_q$ we have $\operatorname{span} \ker f = \operatorname{span}\Gamma_r$ and $\operatorname{span} \ker \hat{f}_q = \operatorname{span}\hat{S}$. As $\hat{S} \subseteq \Gamma_r$, we have $\ker \hat{f}_q = \operatorname{span}\hat{S} \subseteq \operatorname{span}\Gamma_r = \ker f$. By the first isomorphism theorem, we also have $\operatorname{codim} \ker \hat{f}_q = \operatorname{codim} \ker f = d$, as the images of these linear maps span all of $\mathbb{R}^d$. By the third isomorphism theorem we conclude $\Gamma_r = \hat{S}$. Moreover, as $\hat{S} \subseteq \hat{\Gamma}_u \subseteq \Gamma_r$, we have $\hat{S} = \hat{\Gamma}_u = \Gamma_r$.

We now see that $\underline{L}$ is constant on $\Gamma_r$ since there is some link function $\psi : \mathbb{R}^m \to \mathbb{R}$ such that $\Gamma_r = \hat{\Gamma}_u \subseteq \underline{L}_{\psi(u)}$, meaning $\underline{L}(p) = \psi(u)$ for all $p \in \Gamma_r$. This statement contradicts the assumption that $\underline{L}$ is non-constant on $\Gamma_r$. $\qquad \square$

# F  Omitted Discussion and Examples

## F.1  Note on restricting minimizable assumption

While some popular surrogates such as logistic and exponential loss are not minimizable, these losses are still covered in Corollary 3 and Theorem 2 as $\Gamma(p) \neq \emptyset$ when $p \in \mathcal{P} := \operatorname{relint}(\Delta_{\mathcal{Y}})$; moreover, by thresholding $L'(u, y) = \max(L(u, y), \epsilon)$ for sufficiently small $\epsilon > 0$ we can achieve $L' \in \mathcal{L}$ for both. We expect that a generalization of property elicitation which allows for "infinite" predictions (e.g., along a prescribed ray) would allow us to assume minimizability without loss of generality, as convex losses would always admit this more general minimizer.

## F.2  Lower-bounding the convex consistency dimension of the variance

**Corollary 7.** *Let $\mathcal{P}$ be a set of continuous Lebesgue densities on $\mathcal{Y} = \mathbb{R}$ with all $p \in \mathcal{P}$ having the same support. If there exist $p, q, q' \in \mathcal{P}$ with $\mathbb{E}_p Y = \mathbb{E}_q Y \neq \mathbb{E}_{q'} Y$ and $\operatorname{Var}(p) \neq \operatorname{Var}(q)$, then $\operatorname{cons}_{\mathrm{cvx}}(\operatorname{Var}) = \operatorname{elic}_{\mathrm{cvx}}(\operatorname{Var}) = 2$.*

*Proof.* For the upper bound, we may elicit the first two moments via the convex loss $L(r, y) = (r_1 - y)^2 + (r_2 - y^2)^2$, and recover the variance via $\psi(r) = r_2 - r_1^2$, giving $\operatorname{elic}_{\mathrm{cvx}}(\operatorname{Var}) \leq 2$. Now for the lower bound. Without loss of generality, $\mathbb{E}_q Y < \mathbb{E}_{q'} Y$. Let $r = \frac{1}{2}\mathbb{E}_q Y + \frac{1}{2}\mathbb{E}_{q'} Y$, and define $V : \mathcal{Y} \to \mathbb{R}, y \mapsto y - r$. Then $\ker_{\mathcal{P}} V = \{p' \in \mathcal{P} \mid \mathbb{E}_{p'} Y = r\} = \Gamma_r$ where $\Gamma : p' \mapsto \mathbb{E}_{p'} Y$ is the mean. As $\mathbb{E}_q Y < r < \mathbb{E}_{q'} Y$, we conclude $\mathbb{E}_q V < 0 < \mathbb{E}_{q'} V$. We have now satisfied Condition 1 for $d = 1$. To apply Theorem 2, it remains to show that Var is non-constant on $\Gamma_r$. By our assumptions and the definition of Var, we have $\mathbb{E}_p Y^2 \neq \mathbb{E}_q Y^2$. Letting $p_1 = \frac{1}{2}q + \frac{1}{2}q'$, $p_2 = \frac{1}{2}p + \frac{1}{2}q'$, we have $\mathbb{E}_{p_i} Y = r$ for $i \in \{1, 2\}$, but $\mathbb{E}_{p_1} Y^2 = \frac{1}{2}\mathbb{E}_q Y^2 + \frac{1}{2}\mathbb{E}_{q'} Y^2 \neq \frac{1}{2}\mathbb{E}_p Y^2 + \frac{1}{2}\mathbb{E}_{q'} Y^2 = \mathbb{E}_{p_2} Y^2$. As $p_1, p_2$ have the same mean but different second moments, we conclude $\operatorname{Var}(p_1) \neq \operatorname{Var}(p_2)$. $\qquad \square$