# OpenReview forum: "Unifying lower bounds on prediction dimension of convex surrogates"
_NeurIPS.cc/2021/Conference — NeurIPS 2021 Poster_

### Official Review · Reviewer_5yYm · 2021-07-11

**Rating:** 7
**Confidence:** 3

**Summary:**

This paper introduces the notion of d-flats for lower bounding the convex elicitation complexity of a property, which also lower bounds the convex consistency dimension of the property. These quantities are studied in the design and analysis of loss functions to accomplish prediction tasks.

The authors use d-flats to establish lower bounds on the convex elicitation complexity of a variety of properties, such as classification with an abstain option, variance, CVaR, mode, and modal interval.

**Limitations And Societal Impact:**

Yes.

**Main Review:**

The paper is exposited clearly and has a clear narrative: the proposed method of lower bounding convex consistency dimension by using d-flats to lower bound a smaller quantity, convex elicitation complexity, is a useful technique which unifies previous results and provides lower bounds for some properties which were not previously known. The technique itself appears natural and elegant, derived by writing the subgradient condition for optimality and deducing from it using elementary properties that there exists a subspace in the probability set of codimension at most d. The approach demonstrates novelty, as the most prominent previous approach for lower bounding convex consistency dimension was the feasible subspace dimension approach provided by Ramaswamy and Agarwal [33], which the authors show is weaker than the method of d-flats (the d-flat result derives the feasible subspace result in the appendix). Furthermore, the new lower bounds on the complexities of classification with an abstain option, variance, CVaR, mode, and modal interval also demonstrate the novelty.

A question that I have is whether the method of d-flats can be applied to the multiclass version of the classification with abstain option problem; can a corresponding lower bound be proved without drawing the level sets and visually observing that no lines through a point fit in the level set.

Another question is, what is the significance of lower bounding the prediction dimension of a property? Does this have any bearing on how people normally predict properties in practice? Unless stated otherwise, it feels like most of the time, people know what loss function to use to obtain a property, and the lower bound isn't necessarily adding anything to the practice.

(minor typo in line 83, "prediction dimension of d")

**Time Spent Reviewing:**

5

---

> ### Author Response · Authors · 2021-08-10
> **Response to questions**
>
> Thank you for your time and feedback.  On your questions:
>
> 1. The application of the abstain example in Section 3.1, Q1, is for a mutliclass (3-label) version of abstain, so we assume the reviewer means even more labels, $n>3$, where indeed visualization becomes more challenging.  If $\Theta(\log n)$ is the correct convex elicitation complexity for abstain, i.e. if the BEP surrogate achieves the lowest possible prediction dimension, then we believe our bound is not strong enough to match for large $n$.  (It does apply and gives nontrivial lower bounds, but we believe slower than $\log n$.)  If indeed $\log n$ is the correct complexity, we suspect that a stronger version of our bound would suffice, wherein one requires the flats to be “coordinated” in the sense that they must all arise from the *same* convex function.  How to derive a concise, testable condition capturing this cross-flat coordination is a very interesting open question.
>
> 2. See the response to K35K for some comments on the significance of low prediction dimension, and of lower bounding it.  (In short: low prediction dimension can help both computational and statistical efficiency, and lower bounds increase our fundamental understanding of how to design consistent surrogates.)  On the second part of your question, we are motivated by domains where it is not obvious what surrogates could be consistent.  In structured prediction, and even multiclass classification, there is a long history of practitioners using inconsistent surrogates before this fact is discovered.  In these settings, knowledge of what consistent surrogates are possible would save time searching for a surrogate that does not exist.  Another motivating example is the quest for loss functions to evaluate/estimate risk measures, which has been extremely active in the statistics and finance communities.  Lower bounds like the ones we derive similarly have the power to change practice.  As one example, both upper and lower bounds spurred the adoption of loss functions and backtesting procedures such as those proposed by Fissler et al. for CVaR, and could change the way financial institutions are regulated.  In sum, while our main results are impossibilities, they are closely motivated by and have the potential to impact practice.

---

> > ### Comment · Reviewer_5yYm · 2021-09-01
> > **Additional questions about d-flats vs. feasible subspace dimension**
> >
> > I have some additional questions about d-flats vs. the feasible subspace dimension proposed earlier in [33] and I believe also appears in [2] as Theorem 9, as noted in your paper, it would be great if you could respond.
> >
> > 1. Is it true that the proof of the d-flat theorem and the proofs for the feasible subspace theorem are essentially the same, i.e. showing that the probability simplex must contain the null space of a linear map W in d x n, except you conclude the slightly stronger fact that this means the probability simplex contains an entire affine space in a level set, while the previous works simply conclude that the affine dimension of the level set is at least that of the affine space?
> >
> > 2. Which of the results of your paper can and can't be proved by the previous result? I believe that the abstain example (Q1) and hierarchical classification example (Q2) can't, but Q3 (mean) and Q4 (variance) can. What about the others?
> >
> >
> > [33] Harish G Ramaswamy and Shivani Agarwal. Convex calibration dimension for multiclass
> > 450 loss matrices. The Journal of Machine Learning Research, 17(1):397–441, 2016.
> >
> > [2] Arpit Agarwal and Shivani Agarwal. On consistent surrogate risk minimization and
> > 372 property elicitation. In JMLR Workshop and Conference Proceedings, volume 40, pages
> > 373 1–19, 2015. URL http://www.jmlr.org/proceedings/papers/v40/Agarwal15.pdf.

---

> > > ### Author Response · Authors · 2021-09-03
> > > **Question response**
> > >
> > > Thanks for those questions.
> > > 1. Both the proof of [33, Theorem 16] and our d-flat result involve the nullspace of a similar linear map. The phrase “essentially the same” is of course subjective, but we would disagree. We view our approach as offering three key innovations:
> > >     1. a simpler proof via indirect elicitation (the proof of [33, Theorem 16] is several pages, and it further relies on Lemma 23, Lemma 15, and Theorem 7);
> > >     1. a sharper result, by focusing on containment of flats in level sets rather than a more local condition (moreover, [33]’s choice of flat, which focuses on a point p on the boundary of the level set, is sometimes suboptimal when you use the stronger containment condition); and
> > >     1. a more general result, in that it applies to Q2, Q3, Q4.
> > > The reviewer pointed out (2) as the main difference between the two approaches, yet (1) and (3) are also substantial in our opinion.
> > >
> > >
> > > 2. Calibration as defined in [33] does not apply beyond Q1.  While one could extend the definition of calibration to be with respect to a property, and therefore the results in [33] could extend to Q3, how they could extend to Q2 and Q4 is far from clear.  In particular, there has to be a target loss for [33, Theorem 16], meaning that in Q2, Q4, the target statistic has to be elicitable.  This requirement comes from [33, Lemma 23] used to prove [33, Theorem 16] and the characterization of the level sets of elicitable properties forming power diagrams [Lambert et al; 24, 25].  As none of the risk measures in Section 4 are elicitable by a target loss, nor is the variance, the previous result [33] cannot apply to these problems, whereas our more general result can.

---

> > > > ### Comment · Reviewer_5yYm · 2021-09-03
> > > > **Clarification of contribution**
> > > >
> > > > Thanks for the response. I still think the proofs of this paper's Theorem 1, [33, Theorem 16] and [2, Theorem 9] are quite similar; they all say that the level set of a property (for 33, it is the set of probability distributions whose optimal prediction is t, but I think it is not hard to abstract "optimal prediction for a probability" as a "property of probability distribution") contains the intersection of the null space of a particular linear map with the probability simplex. [2, Theorem 9] seems to define the exact same linear map as this paper (columns are subgradients for each label at optimal prediction vector) and [33, Theorem 16] defines basically the same one, it's just generalized to "epsilon-subgradients" because they don't assume existence of minimizers and thus take subgradients w.r.t. a minimizing sequence.
> > > >
> > > > I think the only real difference in the proof is, instead of then concluding that there are at least n - d - 1 (modulo some casework where a probability vector has zero entries) feasible directions in the level set starting from the probability distribution the prediction vector is optimal for, this paper stops at "the level set contains the intersection of the probability simplex with the aforementioned null space." The conclusion of this paper's proof is stronger than the previous ones because it stops and does not try to conclude anything more specific.
> > > >
> > > > This definitely seems useful to point out, though, as a set might contain a line segment but not the intersection of the full extension of that line with a mother set, hence the stronger results for the abstain and hierarchical example. I also really appreciate how much simpler the presentation of this paper is compared to [33] and consider it a valuable contribution.
> > > >
> > > > Also, I think that properties vs. optimal prediction for a probability distribution is a superficial distinction for this set of papers. Doesn't everything still apply with "set of probability distributions for which this is the minimizer" replaced with "set of probability distributions with this as a property"? I have not examined this part carefully, but while [33, Lemma 23] does use the latter as an assumption, it seems only for showing that the feasible subspace dimension for a distribution contained in two level sets is the same with respect to either level set. Not having that doesn't seem to impede in any way the way they lower bound the feasible subspace dimension of any particular level set in [33, Theorem 16].
> > > >
> > > > So I guess my takeaway is that this paper's main contribution is in how it sets up a simple, abstract enough framework for this problem making it good for conveying the subject to readers and a theorem which uses a previous technique, but looks at the result differently and thus gets something simpler and stronger than before. The infinite dimensional results seem interesting and another valuable contribution as well, but I am not sure whether the previous result could not have been used to get the same ones, namely is the global nature of d-flat used for these results.

---

### Official Review · Reviewer_GhRx · 2021-07-15

**Rating:** 6
**Confidence:** 2

**Summary:**

This paper presents a new tool to generate lower bounds on the convex consistency dimension of continuous and discrete prediction tasks. The authors apply this tool, called $d$-flats, to yield new bounds on convex consistency dimension for risk measures, mode and modal interval. They show that d-flats recover and tighten the feasible subspace dimension result in discrete prediction settings.

**Limitations And Societal Impact:**

Yes

**Main Review:**

The new tool used in this paper is based on property elicitation, which is a widely studied area in statistics and economics. Compared with calibration, indirect elicitation is a weaker and simpler condition for consistency. It’s interesting that newer bounds obtained based on indirect elicitation are as tight as calibration-based bounds.

The authors give the first prediction dimension bounds for risk measures with respect to convex surrogates via d-flats.

The authors refer to the four cases–target loss vs. target statistic, and discrete vs. continuous predictions–as the “four quadrans” of supervised learning problems. Using property elicitation, they are able to yield lower bounds on prediction dimension of any convex surrogates. Moreover, these bounds apply across all four quadrants.

This is a well written paper and I recommend its acceptance. The theory in this paper will be helpful to understand how to construct consistent convex surrogates for target losses or properties.

I have some (minor) comments listed below.

1.	This paper considers examples like classification and regression problems, I was wondering if d-flats could be used to study the subset ranking losses.
2.	In the first two paragraphs of the “Setting” section, some notations are not clearly defined, like $\mathcal{B}(\reals^d)$ and $\Delta_{\Y'}$.

**Time Spent Reviewing:**

15 hours

---

> ### Author Response · Authors · 2021-08-10
> **Response to questions**
>
> Thank you for the careful reading and review.  To your questions:
>
> 1. Standard sunset-ranking losses such as Normalized Discounted Cumulative Gain, Mean Average Precision, and Pairwise Disagreement all fall under Quadrant 1, so in that sense our framework applies.  Lower bounds of the convex calibration dimension (equivalent to convex consistency dimension in Q1) for these examples are given in Section 5 of Ramaswamy et al [33] by feasible subspace dimension; we could apply our framework to generalize these results similarly to the generalization of feasible subspace dimension in Section 3.2
>
> 2. Thank you; we will correct these notation omissions.

---

### Official Review · Reviewer_K35K · 2021-07-18

**Rating:** 6
**Confidence:** 1

**Summary:**

The authors introduced a new tool for proving lower bounds on the convex consistency dimension, which is defined as the smallest dimension d such that there exists a link function and convex loss function L of dimension d that are consistent. They showed a connection between the convex consistency dimension and the existence of d-flat. The result can be applied to a wide range of tasks, such as classification, regression and variance estimation. In particular, they apply the tool on risk measure (4.1) and mode and modal interval (4.2) and obtain new lower bounds in these settings.


**Limitations And Societal Impact:**

Yes

**Main Review:**

Questions:
1.What are the practical implications of this result? Will the theory here provide guidance on how to select a surrogate loss function?
2.The paper never defined consistency in the main text despite using the term frequently. I would suggest moving Definition 7 and 8 to the main text.

I apologize that I have no expertise in this area and don’t know anything about the relevant literature, and therefore my confidence is low. The paper seems to provide a nice tool for proving convex consistency dimension lower bound. Though I don’t know about the impact and significance of the result in the area. Therefore I will vote weak accept for now.


**Time Spent Reviewing:**

5

---

> ### Author Response · Authors · 2021-08-10
> **Response to questions**
>
> Thank you for the review and time invested.  In response to your questions:
>
> 1. In several domains, such as structured prediction, it is important to find surrogate losses with low prediction dimension, relative to the often-exponentially-large number of labels.  (For structured problems, a surrogate with low prediction dimension can be necessary for computational efficiency; we conjecture that a low prediction dimension can also improve statistical efficiency, i.e., sample complexity.)  The lower bounds we develop are a key step in a broader research agenda of understanding the design of consistent surrogates -- in particular, they reveal fundamental limitations of surrogate design.  Our bounds also resolve important questions for continuous statistical estimation problems (Sec 4), with applications to financial regulation and other domains, revealing fundamental limitations in these problems as well.
>
> 2. We agree that moving definitions 7 and 8 to the main paper would help matters, as the reviewer suggests; if accepted the extra page will easily accommodate them.

---

### Official Review · Reviewer_kpyj · 2021-07-23

**Rating:** 6
**Confidence:** 1

**Summary:**

This paper looks at convex surrogates of supervised loss functions and tries to obtain lower bounds for the convex consistency dimension which is the lowest prediction dimension with a consistent convex surrogate. They look at two variants of the problem, namely target loss and target statistic and separately for discrete and continuous cases. They define a tool called d-flat which can be used to bound the convex consistency dimension. The work largely relies on the condition of property elicitation, which is a weaker condition than calibration and is defined on a variety of domains such that the convex elicitation complexity lower bounds the convex consistency dimension. They also use the idea to find solutions to two open problems, one related to lower bounds of risk measures and the other for mode and modal interval.

**Main Review:**

This paper looks at convex surrogates of supervised loss functions and tries to obtain lower bounds for the convex consistency dimension which is the lowest prediction dimension with a consistent convex surrogate. They look at two variants of the problem, namely target loss and target statistic and separately for discrete and continuous cases. They define a tool called d-flat which can be used to bound the convex consistency dimension. The work largely relies on the condition of property elicitation, which is a weaker condition than calibration and is defined on a variety of domains such that the convex elicitation complexity lower bounds the convex consistency dimension. They also use the idea to find solutions to two open problems, one related to lower bounds of risk measures and the other for mode and modal interval.

The paper has a lot of involved concepts, particularly for a reader not familiar with the area, but the authors provide enough context and links to related work to form a broad idea of the general principle of both convex consistency dimension and the idea of d-flats/property elicitation. I am also curious whether understanding about the lower bound of the convex consistency dimension necessarily means attainability (the lower bound always is reached for some convex surrogate) and also whether it would be possible to estimate such a surrogate given the understanding of the prediction dimension.

**Time Spent Reviewing:**

2

---

> ### Author Response · Authors · 2021-08-10
> **Attainability of the lower bounds**
>
> Thank you for the thoughtful review.  On whether the lower bounds we present are attained by surrogates, i.e., whether the lower bound is tight, the answer is not always.  (See response to GhRx.)  Our bounds are tight for many problems, though, as is the case for most of our examples and applications.  For example, the bound we present for Q1 in Section 3.1 matches the upper-bound construction of the BEP surrogate from Ramaswamy and Agarwal [33], and the Q4 example in the same section matches known results about the elicitation complexity of the variance.  Our bounds are also tight for entropies and some families of risk measures, though whether our bound is tight for CVaR is open (see conjecture in line 302).

---

### Decision · Program_Chairs · 2021-09-27

**Decision:**

Accept (Poster)

**Comment:**

This paper considers the problem of constructing convex surrogates for supervised learning tasks, and provides lower bounds for the convex consistency dimension, which is the lowest prediction dimension with a consistent convex surrogate. Reviewers felt that paper was well-written, and that the techniques are novel and generally interesting. In particular, the authors use their framework to resolve open questions of Frongillo and Kash (2015) and Dearborn and Frongillo (2020). However, given that the topic is somewhat niche for the NeurIPS community, the authors are encouraged to incorporate the reviewers' suggestions to better convey the significance of their results to the broader community.